# The prognostic effects of somatic mutations in ER-positive breast cancer

Obi L. Griffith [1,2,3,4], Nicholas C. Spies[1], Meenakshi Anurag[5,6], Malachi Griffith[1,2,3,4], Jingqin Luo[3,7], Dongsheng Tu[8], Belinda Yeo[9], Jason Kunisaki[1], Christopher A Miller[1,2], Kilannin Krysiak [1,2], Jasreet Hundal[1], Benjamin J Ainscough [1], Zachary L. Skidmore[1], Katie Campbell[1], Runjun Kumar[2], Catrina Fronick[1], Lisa Cook[1], Jacqueline E. Snider[2], Sherri Davies [2], Shyam M. Kavuri[5,6], Eric C. Chang[5,6], Vincent Magrini[1,4,10], David E. Larson [1], Robert S Fulton[1,4], Shuzhen Liu[8], Samuel Leung[8], David Voduc[8], Ron Bose[2], Mitch Dowsett[9], Richard K. Wilson[1,3,4], Torsten O. Nielsen[8], Elaine R Mardis[1,3,4,10] & Matthew J. Ellis[5,6]

Here we report targeted sequencing of 83 genes using DNA from primary breast cancer samples from 625 postmenopausal (UBC-TAM series) and 328 premenopausal (MA12 trial) hormone receptor-positive (HR+) patients to determine interactions between somatic mutation and prognosis. Independent validation of prognostic interactions was achieved using data from the METABRIC study. Previously established associations between MAP3K1 and PIK3CA mutations with luminal A status/favorable prognosis and TP53 mutations with Luminal B/non-luminal tumors/poor prognosis were observed, validating the methodological approach. In UBC-TAM, *NF1* frame-shift nonsense *(FS/NS)* mutations were also a poor outcome driver that was validated in METABRIC. For MA12, poor outcome associated with PIK3R1 mutation was also reproducible. DDR1 mutations were strongly associated with poor prognosis in UBC-TAM despite stringent false discovery correction ($q = 0.0003$). In conclusion, uncommon recurrent somatic mutations should be further explored to create a more complete explanation of the highly variable outcomes that typifies ER+ breast cancer.

[1] McDonnell Genome Institute, Washington University School of Medicine, St. Louis 63108 MO, USA. [2] Department of Medicine, Division of Oncology, Washington University School of Medicine, St. Louis 63110 MO, USA. [3] Siteman Cancer Center, Washington University School of Medicine, St. Louis 63110 MO, USA. [4] Department of Genetics, Washington University School of Medicine, St. Louis 63110 MO, USA. [5] Lester and Sue Smith Breast Center and Dan L. Duncan Cancer Center, Baylor College of Medicine, Houston 77030 TX, USA. [6] Departments of Medicine and Molecular and Cellular Biology, Baylor College of Medicine, Houston 77030 TX, USA. [7] Division of Biostatistics, Washington University School of Medicine, St. Louis 63110 MO, USA. [8] Genetic Pathology Evaluation Centre, University of British Columbia, Vancouver V6H 3Z6, Canada. [9] Institute of Cancer Research, London SM2 5NG, UK. [10] Present address: Nationwide Children's Hospital and Department of Pediatrics, The Ohio State University College of Medicine, Columbus 43205 OH, USA. These authors contributed equally: Obi L. Griffith, Nicholas C. Spies, Meenakshi Anurag. Correspondence and requests for materials should be addressed to E.R.M. (email: elaine.mardis@nationwidechildrens.org) or to M.J.E. (email: mjellis@bcm.edu)

While recent genomic studies have provided a comprehensive catalog of genes that accumulate somatic point mutations and small insertions/deletions (indels) in estrogen receptor-positive (ER+) breast cancer, there remains considerable uncertainty as to how these newly discovered mutations relate to disease outcomes[1–3]. Most genomic discovery cohorts were neither uniformly treated nor followed long enough. For ER+ disease in particular, prognostic studies require prolonged observation since relapses often occur after 5 years[4]. Uniform treatment was a feature of a whole-genome sequencing study of samples accrued from a neoadjuvant aromatase inhibitor (AI) clinical trial for ER+ clinical stage 2 or 3 disease, although only short-term anti-proliferative response to AI was reported. This investigation identified that mutations in *MAP3K1*, a tumor suppressor gene involved in stress kinase activation, were associated with indolent biological features and low proliferation rates[5]. The resulting hypothesis was that *MAP3K1* mutation would be associated with favorable outcomes. In contrast, *TP53* mutations associated with poor prognosis features and high proliferation rates.

To more comprehensively address the relationships between somatic mutations and outcomes in ER+ breast cancer, we report herein an approach to detect somatic mutations in DNA isolated from formalin fixed tumor blocks that were over 20 years old. After curating existing mutational data from breast cancer genomics discovery studies (Supplementary Data 1), 83 genes were chosen for analysis (Supplementary Table 1). We applied DNA hybrid capture, sequencing and somatic analysis to three ER+ breast cancer discovery cohorts with contrasting clinical characteristics: an older cohort treated with adjuvant tamoxifen and no chemotherapy (UBC-TAM series[6]), a premenopausal cohort uniformly treated with chemotherapy and randomized to tamoxifen versus observation (NCIC-MA12 clinical trial[7]); and a third mixed cohort that was used to expand the mutational landscape analysis (POLAR) (Supplementary Table 2). We report an analytical pipeline to identify somatic variants while compensating for the lack of matched normal DNA, which is generally unavailable in the setting of older formalin-fixed tumor material. Somatic mutations were analyzed for association with standard clinical variables, wherein mutated *TP53* and *MAP3K1* served as a priori hypotheses for poor and good outcome, respectively. Additional objectives were to identify new mutational hotspots, assess interactions with PAM50-based intrinsic subtypes and to determine mutation frequencies for therapeutic targets. Validation was possible by comparing our results to those in cBioPortal where the genes sequenced in the METABRIC cohort overlapped with the 83 genes investigated in the study described herein. We report consistent associations between *NF1* frameshift/nonsense (FS/NS) and non-synonymous *PIK3R1* mutations and poor outcomes that is independent of standard prognostic variables. *DDR1* mutations were also associated with poor outcomes in the UBC-TAM series and significant despite stringent false discovery correction. We conclude that uncommon recurrent somatic mutations should be further explored to create a more complete explanation of the highly variable outcomes that typifies ER+ breast cancer.

## Results

**Sequencing and final study cohorts**. University of British Columbia Tamoxifen Series (UBC-TAM): these cases were drawn from a well-annotated cohort of patients treated with adjuvant tamoxifen without chemotherapy[6]. A total of 625 of 632 (98.8%) patient samples that fully met study criteria passed a minimum sequencing quality cutoff of at least 80% of targeted exonic bases covered at greater than 20× with other quality metrics described

in the supplementary data (Supplementary Figure 1–5 and Supplementary Data 2). Mean depth was correlated with input DNA and negatively correlated with time since diagnosis (approximate age of sample) and duplication rates were negatively correlated with input DNA and positively correlated with sample age. However, despite these trends, overall metrics were excellent with an average of 135.8× coverage and 3.0% duplicate rate despite the generally low input amounts and old sample age. The final patient population had an average age of 67 at diagnosis (range: 40–89+). All were treated with five years of adjuvant tamoxifen, and were primarily postmenopausal, grade 2 or 3 cancers, of ductal histologic subtype (Supplementary Table 2). All were ER+ (>1% cells positive by IHC) and at least 88.6% were clinically HER2- (13/625 unknown). A subset of 463 of these patients had PAM50 subtyping data available from a previous study[6]. The median follow up in the cohort examined was 25 years and one month.

NCIC-MA12 Trial cohort: these cases were drawn from a clinical trial in premenopausal women treated with a standard adjuvant chemotherapy regimen and randomized to tamoxifen versus observation. A total of 459 patient samples passed the minimum sequencing quality threshold (mean coverage: 200×), of which 328 were hormone receptor positive (HR+; >1% cells positive for ER or PR by IHC), and only the HR+ cohort were included here for most analyses. The majority were premenopausal (mean age of 45). All patients received chemotherapy, and 48% were treated with 5 years of adjuvant tamoxifen. A subset of 255 of these patients had PAM50 subtyping data available. The median follow up in the cohort examined was 9.7 years.

POLAR cohort: this patient series was a case-control study of ER+ (>1% cells positive by IHC) breast tumors, 175 of 194 (90.2%) patient samples passed minimum sequencing quality thresholds (mean coverage: ×75). A case was defined as any patient who relapsed during follow-up, and controls were defined as lacking relapse through a similar follow-up duration. Based on these definitions, there were 91 cases and 84 controls. Of the cases, 43 were early relapses (<5 years since diagnosis) and 48 were late relapses (>5 years). Patients were only included if they received adjuvant endocrine therapy, but chemotherapy was not an exclusion criterion, nor was menopausal status. Because the POLAR study was a case-control design, outcome data could not be easily integrated into prognostic analysis. Therefore, these cases were used in the mutation landscape and hotspot analyses only.

Across the three cohorts, there were 1259 patient samples that passed minimum sequencing quality thresholds and 1128 of these were ER+ (UBC-TAM and POLAR) or ER and/or PgR+ (HR+) (MA12).

**Variant calling and filtering**. A total of over 62 million variants were identified in UBC-TAM. After extensive filtering against a set of nearly 70,000 unmatched normal samples and manual review to eliminate common polymorphisms and false positives (see methods), 1991 putative somatic variants were identified (0 to 26 variants per patient). A set of 1693 mutations was defined as the "non-silent" set for further analysis that excluded sequencing variants in splice regions (except proximal splice site), RNA genes (except *MALAT1*), UTRs, introns, and all silent mutations. Finally, a set of 408 frameshift or nonsense mutations was defined. The same filtering method was applied to both the POLAR and MA12 data sets. A total of 540 putative somatic mutations (436 non-silent, 145 FS/NS) were identified in POLAR, and 2104 (1753 non-silent, 610 FS/NS) in MA12. Full details on these variants are included in Supplementary Data 3 and summarized for key genes in Supplementary Figure 6.

**Mutation landscape analysis**. In 1128 samples passing quality control standards, considering only non-silent mutations, 17 genes were mutated at a rate greater than 5%, and 6 at a rate greater than 10%; *PIK3CA* was the only gene mutated in greater than 20% of samples (Fig. 1a). The order from most recurrent to least for the 10 most frequently mutated genes was: *PIK3CA* (41.1%), *TP53* (15.5%), *MLL3* (13.4%), *MAP3K1* (12.0%), CDH1 (10.5%), MALAT1 (10.0%), GATA3 (9.1%), MLL2 (8.7%), ARID1A (7.2%), and BRCA2 (6.6%). This list correlates well with previously reported recurrently mutated genes. For example, the top 4 most significantly mutated (non-silent) genes in the ER+ subset of TCGA breast project[3] were *PIK3CA* (24.0%), *TP53* (14.6%), *GATA3* (8.6%), and *MAP3K1* (6.1%). Considering METABRIC ER+ patients, the most recurrently mutated genes were PIK3CA (~46%), TP53 (~21%), GATA3, MLL3, CDH1, and MAP3K1 (all ~12–14%) demonstrating slightly higher but very similar frequencies. The overall average non-silent mutation frequency was estimated as 1.6 per MB of coding sequence (range: 0.5–5.8 mutations per MB, excluding samples with no mutations called). In order to determine whether mutations in any gene pair were mutually exclusive or co-occurring in this data set, a pair-wise Chi-squared or Fisher's exact test was performed. Mutations in PIK3CA and MAP3K1 were significantly more likely to co-occur (after BH FDR correction) in the TAM dataset, and were near significance in MA12 although not after correction ($p = 0.08$). These results are summarized in Supplementary Data 4.

**Hotspot analysis**. As anticipated[8], mutations in *PIK3CA* at *E542K*, *E545K*, and *H1047R* were highly recurrent in this study with 69/1259 (5.5%) E542K, 104 (8.3%) E545K, and 181 (14.4%) H1047R mutations (Supplementary Figure 6C). Mutations in the ligand-binding domain of *ESR1* (1.1%) were extremely rare[3,9] (Supplementary Figure 6A). To uncover novel hotspots in these data, both Chi-squared and Fisher's exact tests were performed using mutation frequencies from previous sequencing studies as the expected values (see Methods for definition of multi-study MAF file) (Supplementary Table 3). The most notable novel finding was in *CBFB* (Fig. 1b). At least six different genomic alterations were observed in 15 patients (Supplementary Data 3) that affected the donor splice site of exon 2. Manual review of this splice site identified at least two additional patients with evidence for mutations at this location. The predicted effect of these mutations is skipping of exon 2 or alternate donor site usage, each likely resulting in loss-of-function of the *CBFB* protein. Additional splice site mutations were observed at the exon 2, exon 4, and exon 5 acceptor sites of *CBFB*. ErbB2 exhibited the anticipated profile of activating mutations from earlier publications[10].

**Somatic mutation association with PAM50-based intrinsic subtype**. PAM50 intrinsic subtype calls were obtained from previously published analyses to compare to their mutational profiles for UBC-TAM and MA12 (HR+ only) studies. In both studies about half the patients had luminal A tumors. However, the MA12 cohort had a higher proportion of non-luminal subtypes, with 19.8% HER2-E and 6.6% basal and fewer luminal B tumors (25.1% versus 42.4%) (Fig. 2a, b). As expected, patients with the HER2-E intrinsic subtype were enriched for HER2+ve status compared to other subtypes (Fisher's exact test $p < 0.0001$). Of interest, in the HER2-enriched group there were 51 tumors that were not HER2 amplified and of these 4 were HER2 mutant (~8%), indicating that HER2 mutation could be an occasional explanation for a HER2-E subtype assignment in the absence of HER2 amplification. For NF1 FS/NS mutations, there was also a statistically significant association with the HER2-E subtype ($p = 0.002$) (Supplementary Figure 7B and Supplementary Data 5).

Notably NF1 non-silent mutations were enriched in the HER2-E non-HER2 amplified subgroup, where they were present in 8/51 cases (16%). Compared to the frequency in all other subtypes 12/582 (2%), this enrichment was significant (Fishers exact test $p < 0.0001$) (Supplementary Figure 7A left panel). This association could be reproduced in the METABRIC data with an NF1 non-silent mutation incidence in the HER2-E non-HER2 amplified group of 8/80 (10%) versus 35/1283 (3%) in the rest of the subtypes ($p = 0.003$) (Supplementary Figure 7A right panel). Age-density plots by subtype serve to emphasize the large difference in the median age between the two sample cohorts (43 versus 65), and also the influence of age with respect to the intrinsic subtype incidence. Namely, in the older UBC-TAM cohort, an influence of age on intrinsic subtype was not observed (Fig. 2c). In contrast, in the younger MA12 cohort, there is a younger peak incidence with basal-like breast cancer than Luminal A disease (Fig. 2d). Relationships between intrinsic subtype and mutation patterns were also explored, classifying mutation positive status as "non-silent", "missense", nonsense/frame-shift (FS/NS), or FS/NS+ splice site (Supplementary Data 5). The FDR corrected $p$-value ($q$-value) took into account that 83 genes were examined. However, this level of false discovery detection could be viewed as overly conservative in an exploratory analysis. Therefore, any gene mutation with $q$-value association of <0.2 was considered reportable for the purposes of subsequent validation efforts[11–13]. For MA12, non-silent TP53 mutation was highly subtype-associated because of the very high incidence in non-luminal versus luminal subtypes. PIK3CA and MAP3K1 mutations were associated with Luminal A disease in both cohorts (Supplementary Figure 7B). Finally, there was a strong association between Luminal B status and non-silent (Supplementary Figure 7C), as well as FS/NS mutations in GATA3 (Supplementary Data 5, $q$-value = 0.006) for MA12 (but not UBC-TAM). GATA3 mutations were present in 28–30% of Luminal B cases and less so in luminal A cases (5%). Considering $q$ values of <0.2 the associations between FS/NS and non-silent mutations in ATM and Luminal B tumors in MA12 (8–13%) suggests that ATM disruption is also a possible luminal B driver (Supplementary Figure 7C), at least in younger women (MA12). Relationships between age and mutation incidence were therefore also explored (Supplementary Figure 7D), with the finding that both ATM mutation and GATA3 mutations were associated with an earlier age of onset within the luminal B category (Fig. 2e, f). Some ATM mutations are likely to be germline (see discussion below), which could partially explain the association with younger age.

**Survival analysis according to somatic mutation**. For the UBC-TAM Series (Fig. 3a) univariate analysis, favorable prognostic associations for breast-cancer-specific survival (BCSS) were detected for non-silent mutations in *MAP3K1*, *ERBB3*, XBP1, and PIK3CA (Fig. 3b, Supplementary Data 6). Adverse prognostic effects were observed for non-silent mutations in *DDR1* and *TP53*, as well as for frame-shift and nonsense (FS/NS) mutations in NF1. An analysis for recurrence-free survival (RFS) produced similar results, except for ARID1B, which was marginally associated with more favorable outcome. A multivariate Cox model was applied to put each gene in the context of clinical parameters (grade, tumor size, and node status). These analyses indicated that the prognostic effects of non-silent DDR1, PIK3CA, GATA3 FS/NS, TP53, and MAP3K1 mutations were independent of grade and pathological stage (Fig. 3c). Multiple correction testing, yielded DDR1 as the only gene that remained significant with a $q$-value of 0.0003 (Supplementary Data 6). For the MA12 clinical trial cohort (Fig. 4a) we focused on overall survival associations, as this was the primary endpoint of the study and the most robust

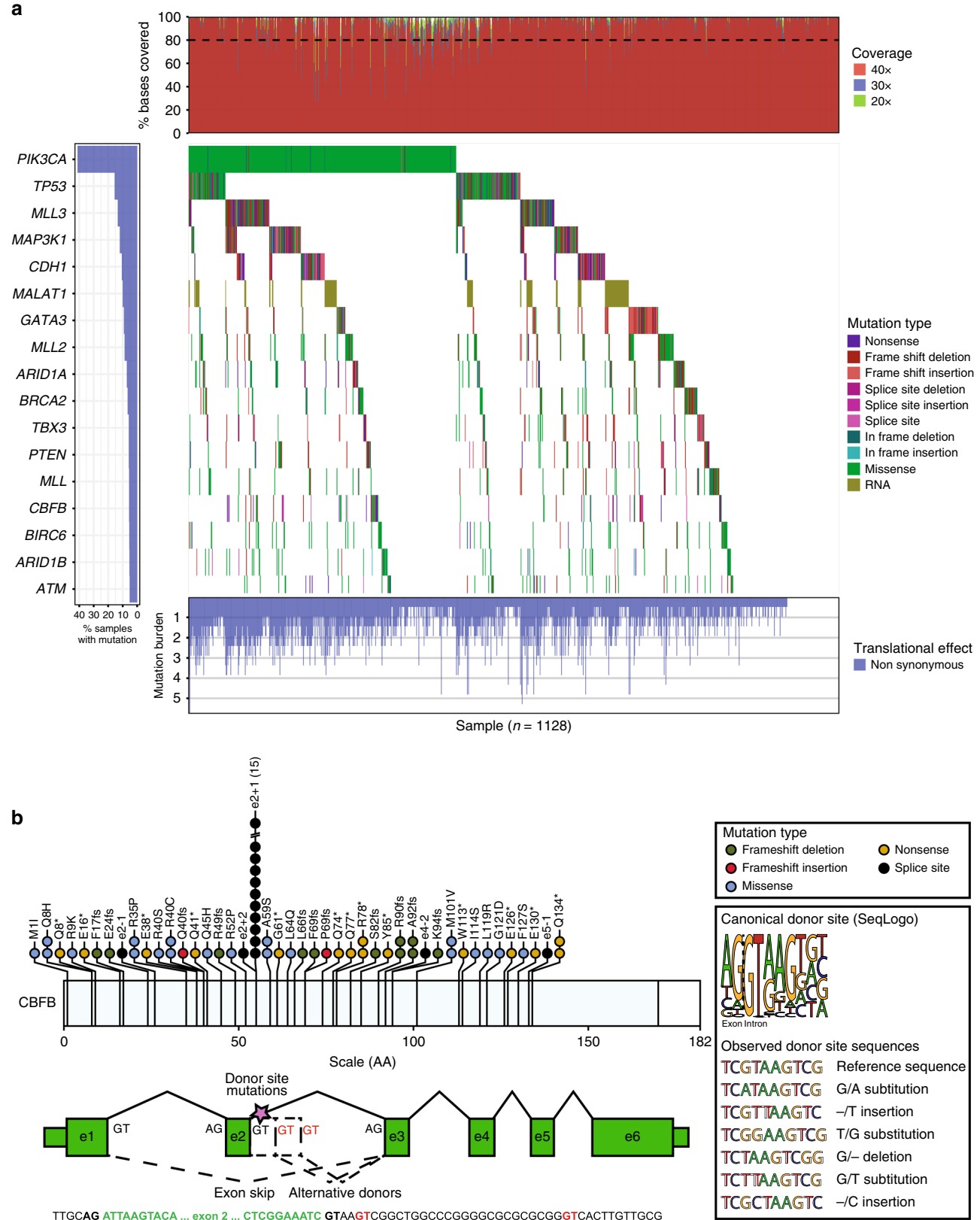

endpoint. A number of rarely mutated genes were associated with poor outcome in univariate analysis as displayed in Fig. 4b. Multiple testing corrections indicated none of these findings could be considered significant[11–13]. However, in multivariate analysis, based on the uncorrected *p*-value, the prognostic effects of mutations in ErbB2, ErbB4, LTK FS/NS, MAP3K4, PIK3R1,

RB1, RELN, and TGFB2 were independent of pathological stage and grade (Fig. 4b).

**Verification of prognostic effects of mutations in METABRIC data.** While few genes were significant in univariate analysis after

**Fig. 1** Mutation recurrence and novel splice site mutation. **a** The overall mutation recurrence rate ranged from 41.1% of samples for *PIK3CA* to 0.0% for *PIN1*. The figure depicts non-silent mutations for all 1128 patients for the top 17 most recurrently mutated genes (>5% recurrence). If a patient had multiple mutations it is colored according to the "most damaging" mutation following the order presented in the Mutation Type legend (vertical color bar). Mutations per MB were calculated using the total number of mutations observed over the total exome space corresponding to the tiled space from "SeqCap EZ Human Exome Library v2.0". A correction factor was applied to account for genes not assayed using the expected number of additional mutations based on ER+ TCGA data. The coverage histogram (top sidebar) shows the percent of targeted exonic bases with at least 20×, 30×, and 40× coverage. **b** Mutation recurrence frequencies (amino acid level) in this study were compared to previously reported mutation frequency from a multi-study MAF file of six reported breast cancer sequencing studies (Supplementary Data 1). An entirely novel mutation "hot spot" was discovered affecting the exon 2 splice (donor) site of *CBFB* in at least 15 patients. Six different single nucleotide substitutions, insertions, and deletions were observed, all affecting either the first or second base of the donor splice site. These mutations were most likely missed in previous studies because of a lack of sequencing coverage due to the GC-rich nature of exons 1 and 2 of *CBFB* (Supplementary Figures 9, 10). Such mutations are predicted to significantly alter the canonical donor site and result in either alternate donor usage or skipping of one or more exons of *CBFB*

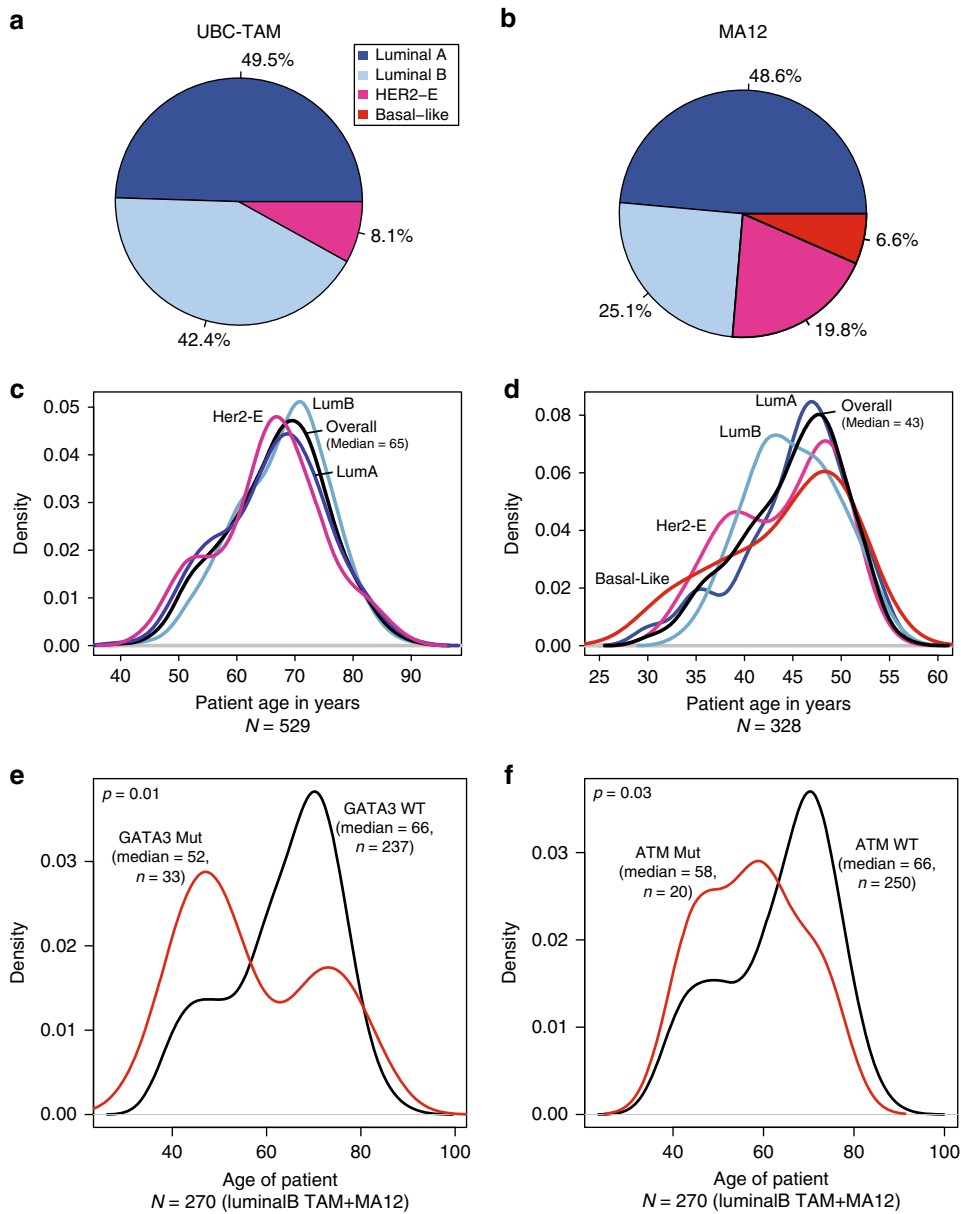

**Fig. 2** Cross-cohort age and subtype analysis. **a**, **b** Percentage composition of samples by intrinsic subtype of the tumor in the two discovery cohorts for UBC-TAM (**a**) and MA12 (**b**) cohorts. **c**, **d** Age-density plots for patients categorized by intrinsic subtype in UBC-TAM (**c**) and MA12 (**d**) cohorts. The overall median age shows that UBC-TAM is constituted mostly of postmenopausal patients (median age = 65), in contrast to MA12, which has younger patients (median age = 43). **e**, **f** Younger luminal B subtype patients harbor GATA3 (**e**) and ATM (**f**) mutations in the combined set of UBC-TAM and MA12 Luminal B cases (median age = 52, $p = 0.01$; median age = 58, $p = 0.03$ for GATA3 and ATM, respectively). Significant $p$ values were determined by Wilcoxon rank-sum test analysis

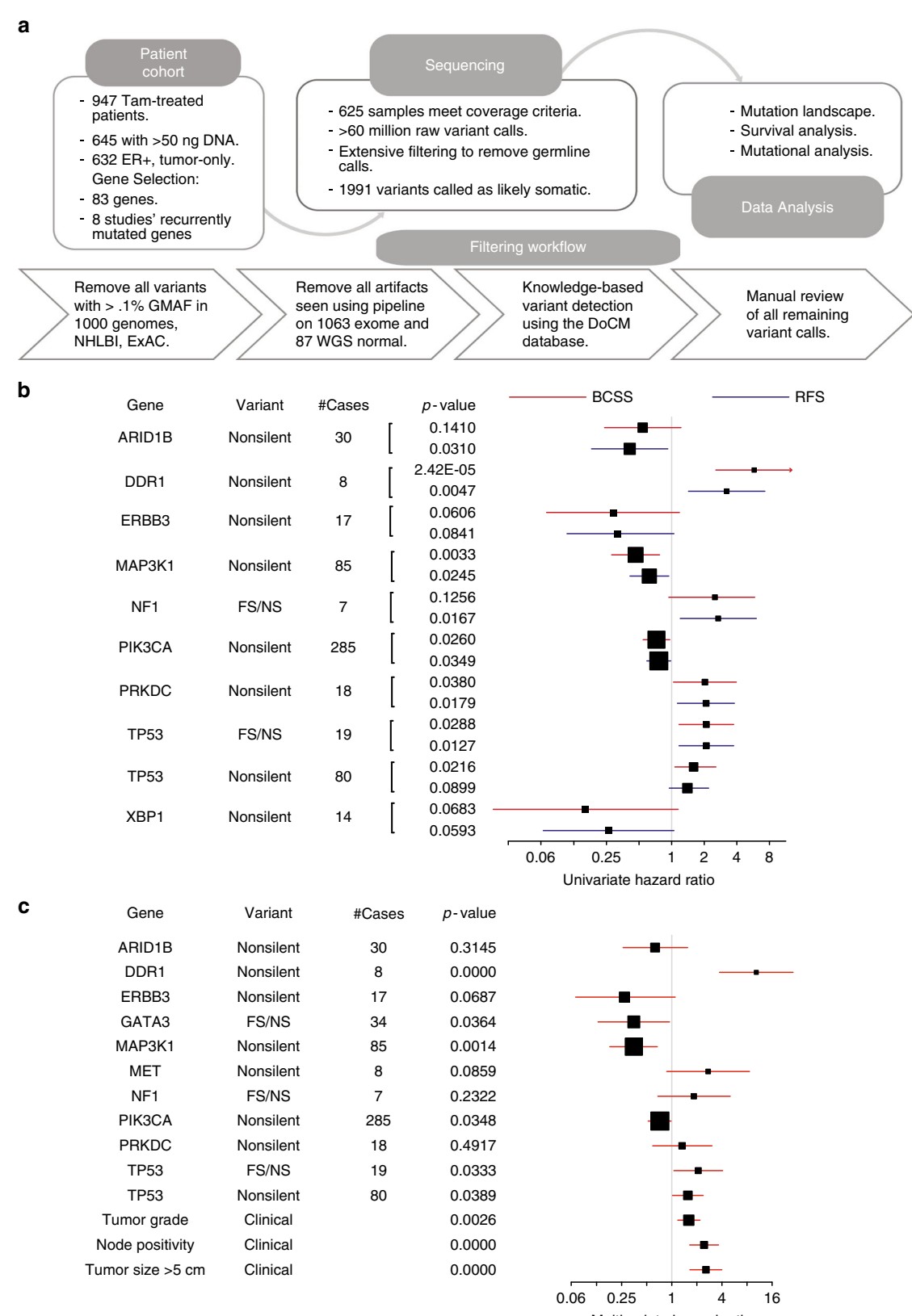

multiple testing correction, their identification provides valuable hypotheses for further testing and validation. We therefore sought additional data in the public domain to further assess the uncorrected *p*-value-based findings in our data set. The METABRIC consortium have reported somatic mutations in cBioPortal[14] with co-reported detailed hormone receptor status, age at diagnosis (median age = 64 years for ER+ patients), mean follow up of >8 years, and disease-specific (breast-cancer-specific) outcome[15,16]. This data set provided the opportunity to conduct a validation exercise for overlapping genes in the two data sets. For

**Fig. 3** Candidate discovery from UBC-TAM cohort and prognosis evaluation. **a** DNA was extracted from tumor specimens from 947 patients with ER+ breast cancer treated with tamoxifen monotherapy for 5 years. 632 samples with adequate yield were sequenced for 83 genes known to be recurrently mutated or breast cancer relevant. A total of 625 samples passed minimum quality checks and were sequenced to an average of 135.8× coverage. A total of ~62 million variants from the reference genome were identified. Extensive filtering and manual review reduced this list to 1991 putatively somatic variants. Survival analysis was applied to non-silent and truncating gene mutation status versus disease outcome (relapse or breast-cancer-specific death). In addition, mutations were analyzed for novel hotspots, patterns of mutual exclusivity or co-occurrence and association with clinical variables. **b** Forest plot of impact of mutations in candidate genes, identified using the UBC-TAM population, on breast-cancer-specific survival (red) and recurrence-free survival (blue). The variant types are characterized based on non-silent or nonsense/frameshift (FS/NS) mutations. The box size is relative to frequency of mutations in the analysis, with larger boxes representing higher incidence mutations. Log-rank test was used to determine significance and Cox regression proportional hazards generated univariate hazard ratios. **c** Multivariate forest plot of effect of mutations in UBC-TAM candidate genes on breast-cancer-specific survival when assessed together with clinical factors including Tumor Grade, Node positivity, and Tumor Size (>5 cm). Multivariate Cox proportional-hazard model was used in the multivariate analysis

the UBC-TAM series (Fig. 3), nine genes with a univariate $p$-value of <0.05 were brought forward for validation (Fig. 5). Of the six overlapping genes also examined in METABRIC, consistent prognostic effects independent of clinical variables were observed for non-silent mutations in three genes, *MAP3K1* (favorable), *TP53* (unfavorable), and *NF1* FS/NS mutations (unfavorable). In order to maintain coherence in discovery and validation patient cohorts, a similar analysis was carried out restricting the patient pool to postmenopausal patients only. No significant variation in hazard ratio for candidate genes was observed (Supplementary Table 4). For the MA12 series (Fig. 4), five shared genes were identified with univariate $p$ values of <0.05, yet only *PIK3R1* mutations (non-silent or FS/NS) showed consistent adverse prognostic effects (Fig. 6). The Kaplan–Meier survival plots for the consistent adverse prognostic effects of *NF1* FS/NS (TAM vs METABRIC) and non-silent *PIK3R1* (MA12 vs METABRIC) mutations are illustrated in Fig. 7a–d. Copy number aberrations and chromosomal instability have been associated with prognosis across multiple cancer types, including ER-positive (ER+) breast cancer[15,17,18]. To gauge the confounding nature of commonly amplified genes in breast cancer, we further performed multivariate analysis on the candidate genes with cases of amplification of MYC, FGFR1, CCND1, and ERBB2 (Supplementary Table 5). We did not observe a significant change in the hazard ratios reported in Figs. 5b and 6b.

**Prognostic interactions between PIK3CA and MAP3K1.** Since PIK3CA and MAP3K1 mutations co-associate, the combined effect of non-silent mutations in these genes was examined. Patients with tumors exhibiting both genes mutated have a more favorable clinical course than either singly mutant cases or cases without either gene mutated. While the prognostic effects were strongest in the UBC-TAM series, this result was also reproduced in the METABRIC data (Fig. 7e, f).

**Mutation analyses for uncommon targetable kinases.** Of the 83 genes analyzed, at least eight are directly targetable with small molecules or antibodies that are either FDA approved or in late-stage development (Fig. 8). Pre-existing data on these mutations is summarized (Supplementary Data 7). PIK3CA is not further discussed here, since the mutation spectrum is well described and large therapeutic studies are already underway. An examination of the 23 mutations in ERbB2 revealed locations that were, as expected, clustered in two major domains, with 2 of 23 having extracellular domain mutations at residue 310 and 21 of 23 having kinase domain mutations between residues 755–842[10,19]. To further investigate the preliminary finding of an adverse prognostic effect for ERBB2 mutation in the MA12 series, an examination of the METABRIC data indicated that known activating mutations in ErbB2 were associated with a near significant adverse effect (HR = 1.71, $p = 0.075$) (Supplementary Figure 8).

For ERBB3, two known activating mutations were identified (V104L and E928A)[20]. In total, ErbB2 exhibited the anticipated profile of activating mutations from earlier publications[10] with 22/1259 (1.7%) samples harboring known activating mutations and another six variants of unknown significance in the kinase domain or at the S310 residue (Fig. 8c). The DDR1 kinase domain mutation, R776W, is possibly homologous to EGFR hotspot mutation L858R, but the remaining DDR1 variants are of unknown significance. For the mutations in JAK1, 3 of 12 are loss-of-function mutations (frame shift or nonsense) and the S816* mutation has been reported in a lung adenocarcinoma sequencing data set[21]. The loss-of-function mutations in JAK1 have been shown to associate with immunotherapy resistance[22,23]. A few mutations identified in ERBB4, MET, and PDGFRA have been previously reported but those reported here have not been functionally tested.

## Discussion

This investigation adds to the knowledge on the impact of somatic mutations in ER+ breast cancer by providing new data on almost a thousand samples with prolonged follow up and controlled adjuvant treatment. By combining these data with a comprehensive overview of published information we report herein the largest database studied to date. The size of our database is critical to our conclusions as the investigation of the prognostic effects of rarer mutations with an overall frequency of 5% or less requires very large sample sizes. Weaknesses include the lack of treatment prediction because endocrine treatment in UBC-Tam was uniform but not randomized. In MA12 the use of tamoxifen was randomized, but the numbers were too small to examine treatment interactions. The landscape of recurrently mutated genes in ER+ breast cancer observed in this study is consistent with reports where matched germline samples were available, indicating that our variant filters were effective for somatic mutation detection in a research setting. Overall, mutation frequencies were higher in our cohort (e.g., for *PIK3CA*, *MLL3*, *MAP3K1*) than the TCGA cohort, but were also lower for a few specific genes (e.g., *TP53* and *GATA3*). Due to higher sequencing data coverage of recurrently mutated target genes than TCGA and the use of a different hybrid capture reagent, we were likely able to detect mutations that were missed with lower-depth exome or whole-genome sequencing data. Differences in patient populations may also be a factor. Frequencies were much closer to reported values for METABRIC, which also used a targeted sequencing approach. It is also possible that in some instances we overestimated somatic mutation frequency, due to the lack of matched normal samples and imperfections in our germline polymorphism filtering. In particular, a significant number of *BRCA1*, *BRCA2*, and *ATM* mutations are likely de novo germline mutations that we would not be able to easily distinguish from somatic mutations. Of the 117 non-silent

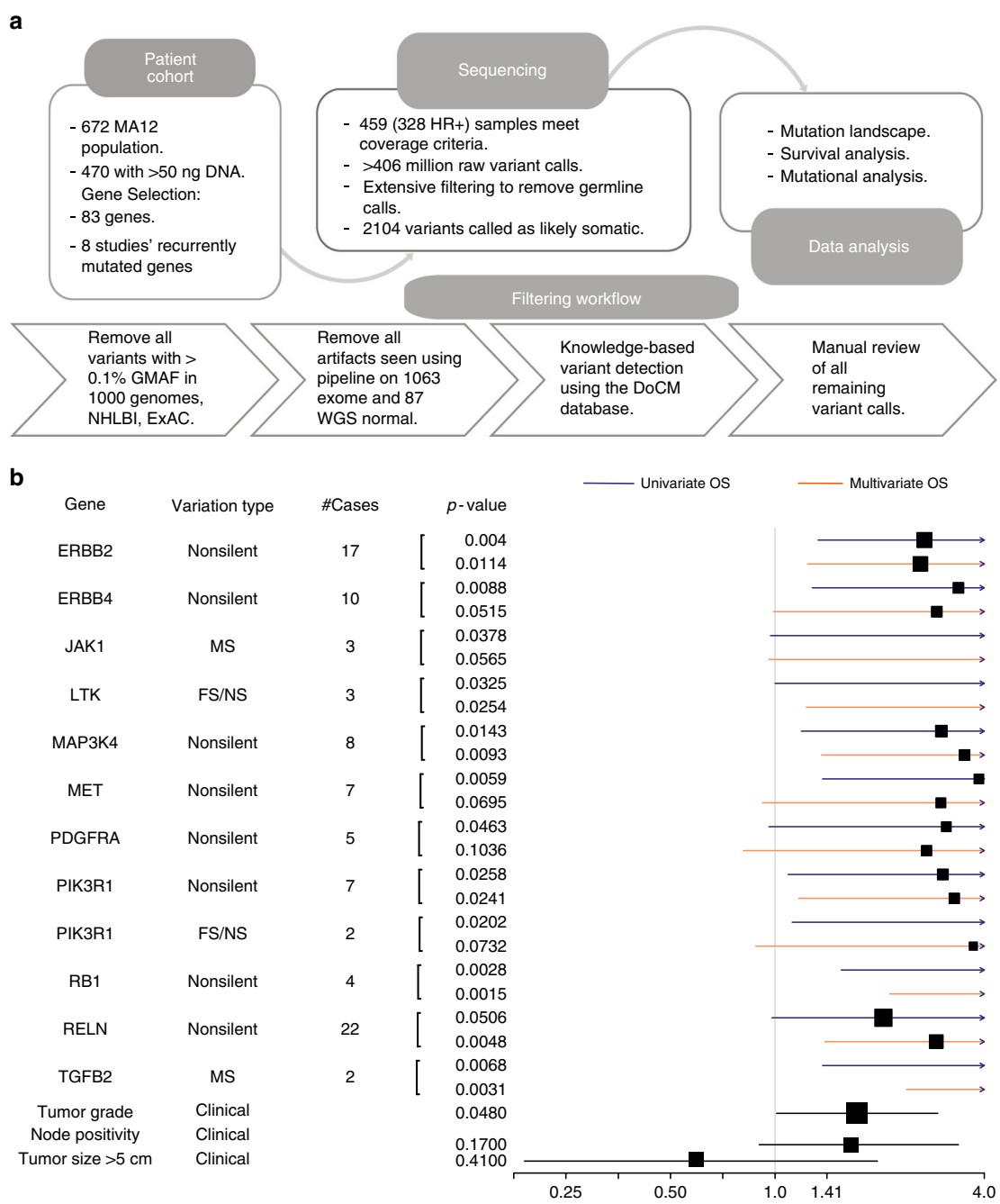

**Fig. 4** Candidate discovery from MA12 cohort and prognosis evaluation. **a** DNA was extracted from tumor specimens and 470 samples with adequate yield were sequenced for 83 genes known to be recurrently mutated or breast cancer relevant. A total of 459 (328 HR+) samples passed minimum quality checks and were sequenced to an average of 272.6× coverage. A total of 406 million variants from the reference genome were identified. Extensive filtering and manual review reduced this list to 2104 putatively somatic variants. Survival analysis was applied to non-silent and truncating gene mutation status versus overall survival. **b** Forest plot showing effect of mutation in candidate genes on overall survival (univariate—blue, multivariate—orange), along with the clinical factors used in the multivariate analysis (black), tumor grade, node positivity, and tumor size (>5 cm). The box size is relative to frequency of mutations in the analysis, with larger boxes representing higher incidence mutations. Note: a few boxes are not shown if their hazard ratios were greater than 4.0. Log-rank test was used to determine significance and Cox regression proportional hazards generated univariate hazard ratios. Multivariate Cox proportional-hazard model was used in the multivariate analysis

*BRCA1/2* mutations observed (from 110/1128 patients across all 3 cohorts; 7 patients had two hits) 74 were observed at a VAF greater than 40% and 31 were greater 60%. Additionally, of the 61 non-silent *ATM* mutations (from 58/1128 samples; 3 samples had 2 hits) 39 had VAF greater than 40% and 18 had VAF greater than 60 (Supplementary Data 9). Variants with VAFs this high

are less likely to be somatic given the general expectation of impure tumor samples and heterozygous mutations. Indeed, the VAFs for *BRCA1/2* and *ATM* non-silent mutations (mean = 46.0%) were significantly higher than for other genes (mean = 36.7%, $p = 5.92e-09$). Even when considered separately, the VAFs for *BRCA1* (mean = 46.6%), *BRCA2* (mean = 43.8%), and *ATM*

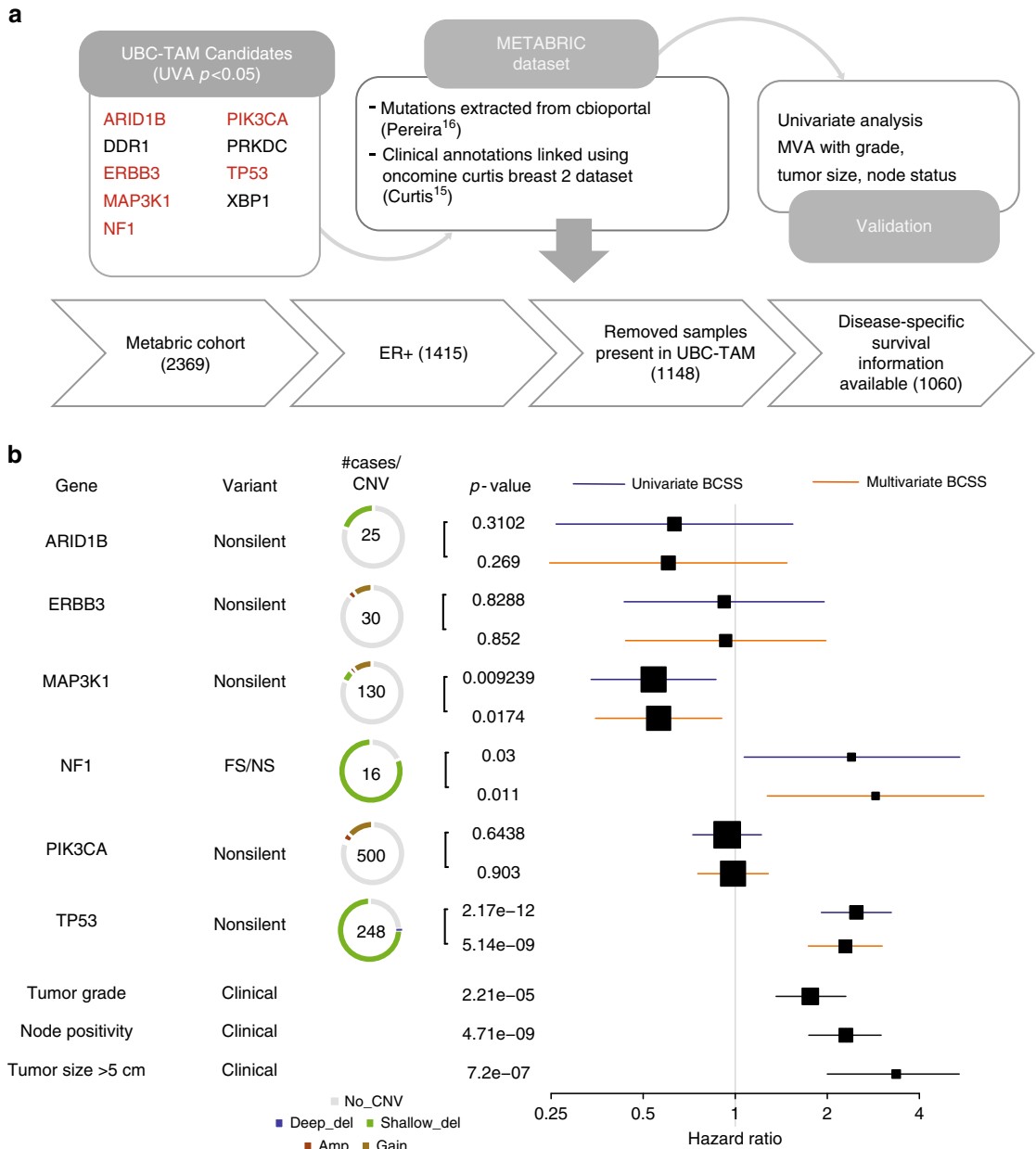

**Fig. 5** Validation of UBC-TAM candidates in ER+ METABRIC. **a** Six out of nine candidate genes from UBC-TAM analysis had mutations reported in the METABRIC cohort. 1060 ER+ samples with breast-cancer-specific survival information were used to test the effect of mutations in the candidate genes on prognosis. **b** Forest plot shows effect of mutated candidate genes on breast-cancer-specific survival in METABRIC ER+ cohort with univariate cox proportional-hazard ratio in blue and multivariate in orange. The clinical factors used in the multivariate analysis, namely tumor grade, node positivity and tumor size (>5 cm), are shown in black. The box size is relative to frequency of mutations in the analysis, with larger boxes representing higher incidence mutations. The # cases/CNV column shows the total number of cases with the SNV/Indel variant surrounded by a ring chart indicating the proportion of total cases with CNV alterations. Log-rank test was used to determine significance and Cox regression proportional hazards generated univariate hazard ratios. Multivariate Cox proportional-hazard model was used in the multivariate analysis

(mean = 48.2%) were significantly higher than the other genes ($p$ = 0.002, $p$ = 0.0015, and 5.27e-5, respectively). Among the *BRCA1/2* variants, there were eight known pathogenic (ENIGMA expert reviewed) mutations according to a search of the BRCA Exchange database (http://brcaexchange.org, 12 November 2017) and another 37 assumed pathogenic (FS/NS) mutations. Of the remaining, four were benign according to expert review (ENIGMA), and eight benign, 15 likely benign and 45 variants of unknown significance according to all public sources. Out of the 61 *ATM* variants queried in ClinVar, four were designated as pathogenic, three were pathogenic/likely pathogenic, and two

were likely pathogenic. Another seven were frameshift mutations and assumed pathogenic. Additionally, 23 variants had uncertain significance, eight variants had conflicting interpretations of pathogenicity (any combination of benign, likely benign, or uncertain significance), and the remaining 14 variants had no data. *ATM* variants were also queried in the Leiden Open Variation Database (LOVD)[24], which identified one variant that affects function (designated as likely pathogenic by ClinVar), 10 variants with unknown effect, and one variant that probably does not affect function (uncertain significance in ClinVar). The remaining variants had no data in LOVD. Given these

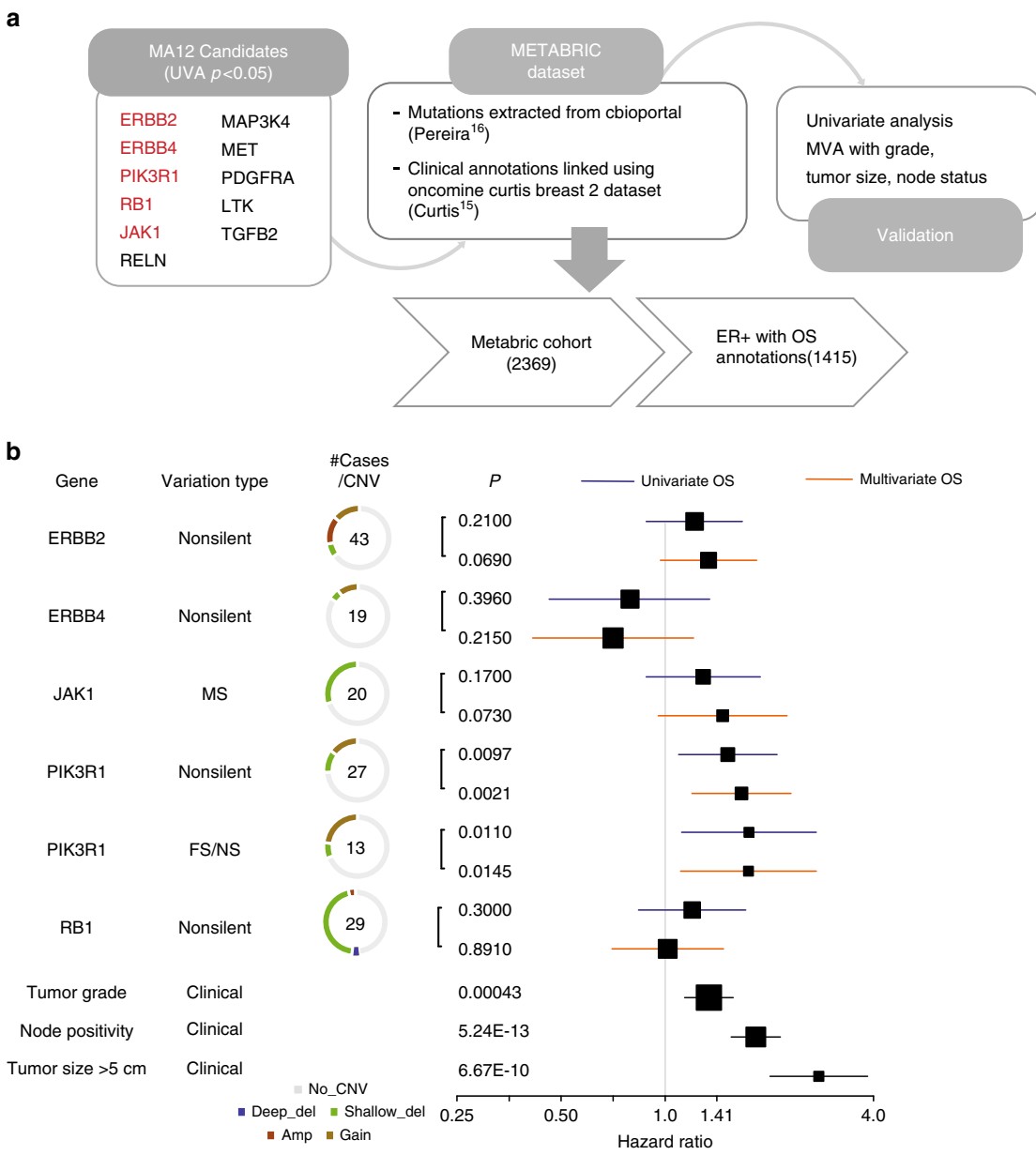

**Fig. 6** Validation of MA12 candidates in ER+ METABRIC. **a** Five out of eleven candidates from MA12 analysis had mutations reported in the METABRIC cohort. 1415 ER+ samples with overall survival information were used to test the effect of mutations in the candidate genes on prognosis. **b** Forest plot shows effect of mutated candidate genes, shortlisted based on MA12 mutation analysis, on overall survival in METABRIC ER+ breast cancer patients. Univariate (blue) and multivariate (orange) cox proportional-hazard ratios depict the independent prediction of survival outcomes for the six candidate genes. The box size is relative to frequency of mutations in the analysis, with larger boxes representing higher incidence mutations. The # cases/CNV column shows the total number of cases with the SNV/Indel variant surrounded by a ring chart indicating the proportion of total cases with CNV alterations Log-rank test was used to determine significance and Cox regression proportional hazards generated univariate hazard ratios. Multivariate Cox proportional-hazard model was used in the multivariate analysis

complexities the prognostic effects of somatic versus germline BRCA1/2 and ATM mutations remain unresolved, however attention should clearly be paid to therapeutic strategies for these patients. The ATM findings deserve a particular highlight because of the younger age/luminal B association and the current lack of studies devoted to this population.

The discovery of a novel recurrent *CBFB* (core binding factor subunit beta) splice site mutation in this cohort illustrates a limitation of exome capture reagents. The affected bases in exon 2 of CBFB display reduced sequence coverage, possibly due to high GC content, in the breast TCGA exome dataset (Supplementary Figures 9–10). This site was mutated in at least 1.5% of ER+

breast cancers sequenced, bringing the overall rate of CBFB mutations to nearly 6%, which should drive further investigation of this gene in ER+ breast cancer pathogenesis. *CBFB* functions as a subunit in a heterodimeric core binding transcription factor that interacts with *RUNX1*[25]. Consistent with this model, *CBFB* mutants were mutually exclusive from *RUNX1* mutants in this cohort with only a single sample harboring non-silent mutations in both *CBFB* and *RUNX1*.

The UBC-TAM and MA12 studies revealed different lists of potentially prognostic mutations. Prognostic effects are likely to be strongly affected by the use of systemic therapy, as well as by patient age at diagnosis. The UBC-TAM series is the simplest

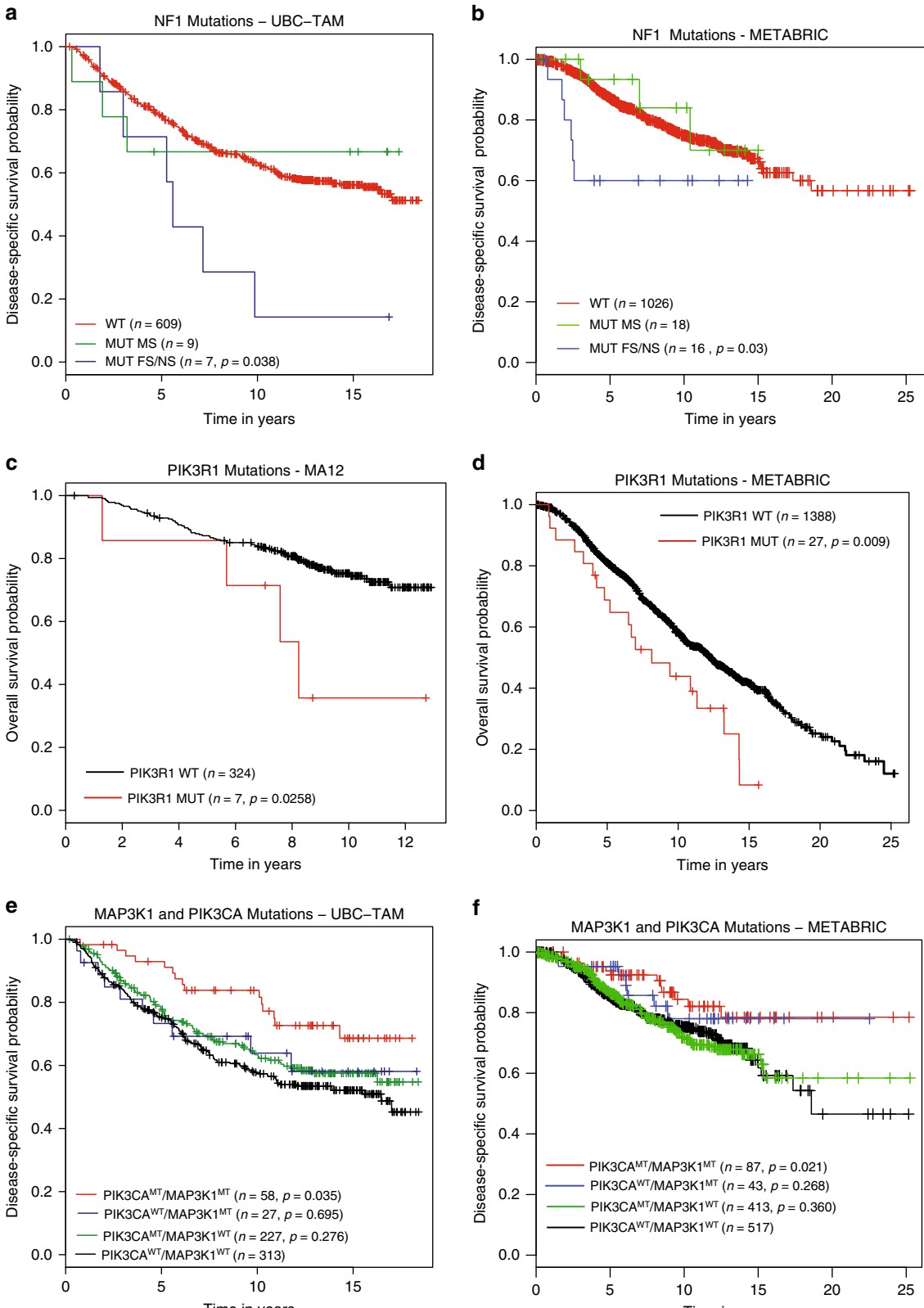

**Fig. 7** Kaplan–Meier plots of candidate gene mutations in discovery and validation cohorts. **a**, **b** Kaplan–Meier graphs showing the prognostic role of NF1 mutations, separated by variant type—Missense (MUT MS, green), Frameshift/Nonsense (MUT FS/NS, blue) in ER+ breast cancer patients from **a** UBC-TAM and **b** METABRIC cohort establishing the association between FS/NS mutations in NF1 with poor prognosis. **c**, **d** Kaplan–Meier graph showing the prognostic role of PIK3R1 in **c** MA12 and **d** METABRIC ER+ breast cancer patients, categorized based on tumors with wild type (WT, black) or mutated PIK3R1 non-silent mutations (MUT, red). **e**–**f** Kaplan–Meier graph demonstrating co-occurrence of non-silent mutations in MAP3K1 and PIK3CA (red) in **e** UBC-TAM and **f** METABRIC associates with better survival when compared against tumors with mutations exclusively in MAP3K1 (blue) or PIK3CA (green) or wild type for both MAP3K1 and PIK3CA (black). *p*, log rank test *p*-value

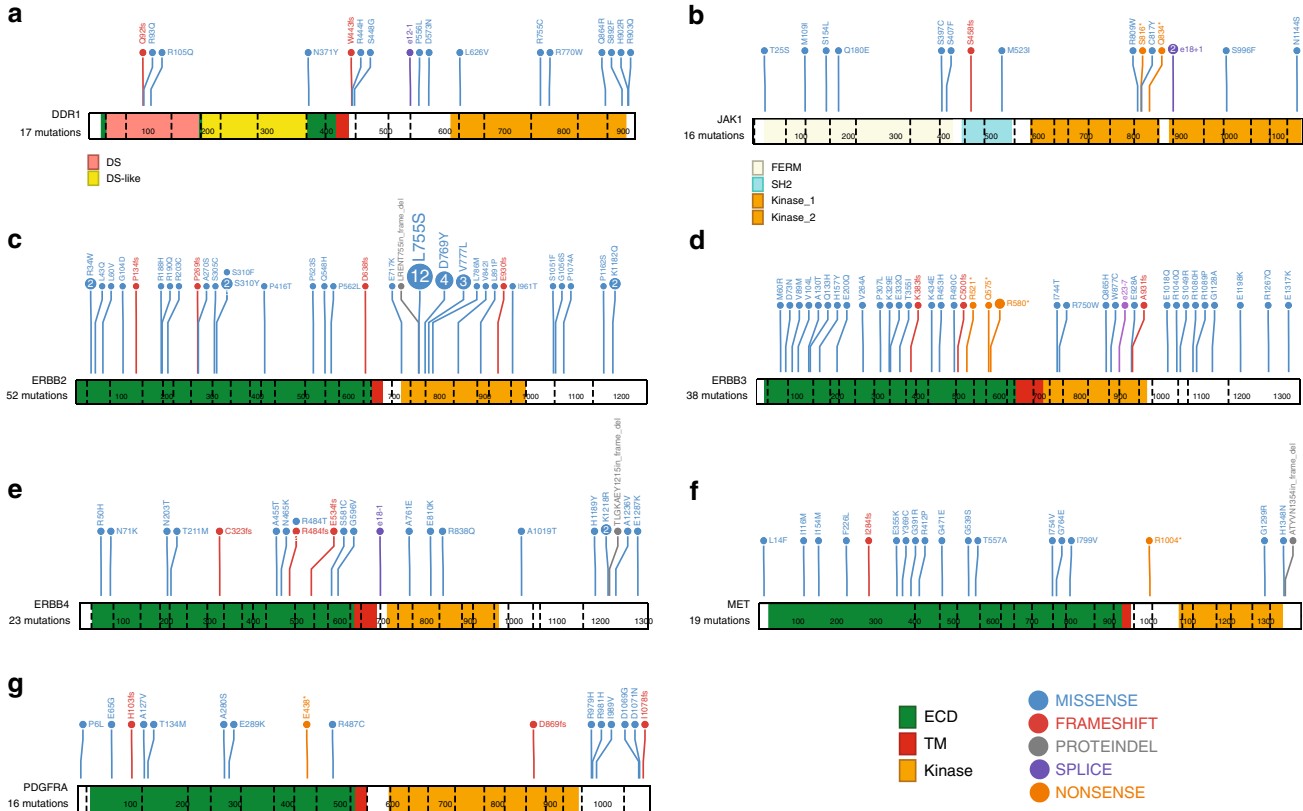

**Fig. 8** Mutation profiles for selected genes. Mutation frequency plots illustrate all non-silent mutations (TAM, POLAR, and MA12; $n = 1259$) for representative transcripts for several kinase genes of interest. The domains belonging to **a** DDR1 (RefSeq ID: NM_013994) and **b** JAK1 (NM_002227) are indicated below the schematic diagram of each gene. The ECD (extracellular domain), TM (transmembrane domain), and kinase domain are depicted as green, red, and orange bars respectively for **c** ERBB2 (NM_004448), **d** ERBB3 (NM_001982), **e** ERBB4 (NM_005235), **f** MET (NM_000245), and **g** PDGFRA (NM_006206). The variant counts across the three data sets for each gene are provided below the gene's name. Note, in the mapping from Ensembl (Supplementary Data 3) to RefSeq annotations (required for use of ProteinPaint tool) a small number of variants annotations may have changed or been lost, despite selecting the most similar representative transcript possible

study to interpret from a drug resistance perspective since the only systemic therapy was tamoxifen. Thus, the consistent adverse effect of NF1 FS/NS mutation on prognosis is intriguing as this result is consistent with results from an in vitro screen for tamoxifen resistance[26]. Understanding why only FS/NS mutations predict poor outcome, rather than missense or other non-silent mutations, will require further investigation. The association of the HER2-E, non-HER2 amplified subset with non-synonymous NF1 mutations was observed in both the discovery and validation (METABRIC) data sets. It is a logical proposition that mutations that activate RAS, like NF1 mutation, could create a tumor with a similar transcriptional phenotype as some HER2 amplified breast cancers. PIK3R1 mutation also emerged as a consistent poor prognosis mutation from the MA12 analysis, with validation in METABRIC. The proposed favorable prognostic effects of PIK3CA mutation were observed in the UBC-TAM series, but were not found to be independent of stage and grade, and PTEN mutations were neutral.

According to our validation results, NF1, PIK3R1, PIK3CA, and TP53 are therefore likely to be prognostic drivers that are independent of clinical variables. In postmenopausal women treated with adjuvant endocrine therapy, DDR1, PRKDC, and XBP1 should be further studied and of these DDR1 is the strongest candidate because it was significant despite strict false discovery correction. DDR1 is a collagen-binding receptor expressed in epithelial cells that stabilizes E-cadherin-mediated intracellular adhesion[27]. DDR1 mutations also occur in

endometrial cancer[28], acute leukemia[29], and lung cancer[30]. Loss of DDR1 (DDR1-null mice) produces hyper-proliferation and abnormal branching of mammary ducts, suggesting DDR1 is a breast tumor suppressor[31]. Mutations in PRKDC will potentially produce a defective ATM response/low ATM levels[32], which is interesting in the context of the finding herein that ATM mutations are a potential luminal B driver gene. The significance of a defective ATM pathway as a cause of endocrine resistance is highlighted by the recent finding that dysregulation of the MutL complex (MLH1, PMS1, and PMS2) causes failure of ATM/CHK2-based negative regulation of CDK4/6[33]. Prognostic candidate mutations revealed by the MA12 analysis were different from the UBC-TAM series, likely reflecting the different patient profiles and adjuvant treatments illustrated in Fig. 2. The prognostic effects of mutations in ERBB2, ERBB4, JAK1, LTK, MAP3K4, MET, PDGFRA, RB1, RELN, and TGFB2 all await further study with even larger sample sizes.

A limitation of this study is that the mutation data sets we generated for UBC-TAM and MA12 cohorts lack comprehensive assessment of copy number signatures that have been associated with prognosis in ER+ breast cancer[15,17,18]. While multivariate analysis considering key CNVs did not appear to affect our prognostic associations, future studies may be needed to completely understand the interplay between simple and large-scale variation for prognostic prediction. Another limitation to this study was the heterogeneity in the data sets in terms of age, treatment, and other factors that limited direct comparison and

made validation with METABRIC somewhat challenging. The collection of sufficiently large, uniformly treated populations with long-term follow-up for discovery and validation remains a challenge that must be addressed to fully characterize the prognostic significance of somatic mutations, especially low frequency mutations.

In conclusion, we have successfully utilized clinically well-annotated, uniformly treated patient samples using DNA from archival material greater than 20 years old without a matched normal to explore the prognostic effects encoded by the mutational landscape of ER+ breast cancer. We were able to confirm our prospective hypothesis from our earlier studies[5] that MAP3K1 is associated with indolent disease and TP53 with adverse outcomes. We also associated NF1 FS/NS mutations with strong adverse effects on prognosis. Similarly, PIK3R1 mutations were associated with an adverse prognosis, in contrast to PIK3CA mutation which were weakly favorable. This suggests somatic mutations in these two physically interacting gene products are not biologically equivalent with respect to PI3 kinase pathway activation and resistance effects. The possibility that the long tail of low frequency mutation events in luminal type breast cancer may harbor multiple molecular explanations for poor outcomes should spur new collaborative efforts to thoroughly screen thousands of properly annotated cases. Only after these iterative efforts of proposing and confirming candidates will a clinically useful and comprehensive somatic mutation-based classification of ER+ breast cancer emerge. In the meantime, functional studies should be pursued to understand the biological effects of low frequency somatic mutations, prioritizing these studies according to whether the mutations are driving an adverse prognostic effect and whether their disruption creates a therapeutic vulnerability.

## Methods

**Patient samples**. The focus of this analysis was ER-positive breast cancer patients with long-term follow-up data available. As such, three main cohorts were available to us, the University of British Columbia Tamoxifen Cohort (UBC-TAM) cohort, the NCIC-MA12 trial (NCT00002542) cohort, and the POLAR cohort. Collection for each of these cohorts is described below. Ethics board approvals were obtained for all studies. For the UBC-TAM series, biomarker studies on the anonymized archival specimens and clinical data were approved by the Clinical Research Ethics Board of the British Columbia Cancer Agency. The MA12 trial was approved by the NCIC Clinical Trials Group (CTG). The details of the conduct of this study are published[34]. Analysis on anonymized archival specimens and clinical data were approved by the Clinical Research Ethics Board of the British Columbia Cancer Agency. For the analysis on the POLAR study, biomarker studies on the anonymized archival specimens and clinical data were approved by the Clinical Research Committee, Royal Marsden Trust and the National Research Ethics Committee Research Ethics Committee, West Midlands, United Kingdom. The Human Research Protection Office of Washington University also approved these anonymized analyses.

UBC-TAM—In total, 947 samples were identified from the TAM series[6,35] with sufficient materials available for DNA isolation. Archival formalin-fixed paraffin embedded (FFPE) blocks were subjected to histopathologic review and one to five cores (1.25 mm) taken from tumor rich areas (>50%) for DNA isolation. Briefly, the cores were de-paraffinized using Citri-Solv Clearing Agent (Fisherbrand # 22–143–975), the pellet washed with 100% ethanol and then dried at 55 °C. The residual tissue pellet was incubated overnight with proteinase K solution (40 μl) and the nucleic acid extracted per manufacturer's directions using the Qiagen Blood and Tissue Kit (catalog #69506). A total of 645 samples met the minimum DNA yield requirement of 50 ng. Of these, five samples were excluded as duplicates (higher yield sample was chosen per patient). An additional eight patients were excluded if they had a metastasis at diagnosis or were not treated with tamoxifen upon review of records. A minimum sequencing quality cutoff was applied, requiring at least 80% of targeted bases covered at greater than 20× with other quality metrics described in the supplementary data. The final population that met all study criteria and had sufficient coverage (See REMARKS summary in Fig. 3a for details) included 625 women diagnosed with estrogen receptor-positive invasive breast cancer in the province of British Columbia of Canada between 1986 and 1992, as previously described[6,35–37]. These patients were treated with five years of tamoxifen as the adjuvant monotherapy. The mean age of the cohort at diagnosis of breast cancer was 66.9 years, and the median follow-up time was 11.7 years.

MA12—DNA isolations were performed as above. A total of 459 of 470 (97.7%) patient samples met study criteria and passed the minimum sequencing quality cutoff. The final patient population for most analyses only includes 328 HR positive patients (see REMARKS summary in Fig. 4a for details), who had an average age of 45 at randomization (range: 30–57). All received chemotherapy (27% CEF, 44% CMF, 29% AC) and 48% were treated with five years of adjuvant tamoxifen. Patients were primarily premenopausal, with grade 2 or 3 (63%), nodal positive (82%), and stage 2 (85%) breast cancers. 89% were clinically ER+ and 53% clinically PR+ (42% unknown). PAM50 molecular subtype by a non-commercial qRT-PCR assay was available from a previous study on 255 (78%) patients: 114 (44.7%) were luminal A, 61 (23.9%) luminal B, 48 (18.8%) Her2-enriched, 17 (6.7%) normal-like, and 15 (5.9%) basal-like subtype.

POLAR—DNA was isolated as above from 196 primary breast tumors from the POLAR study, a case-control study of ER+ primary breast tumors from patients diagnosed with breast cancer between 2000 and 2004 (to allow for long-term follow up). A case was defined as a patient who relapsed with any recurrence during follow-up. A control was defined as a patient who did not relapse during follow-up. Patients were included if they were treated with 5 years of adjuvant endocrine therapy. Pre and postmenopausal patients were eligible for the study and chemotherapy was also permitted. There are three main differences between the TAM series and the POLAR case-control cohort that warrant attention.

1. TAM had no patients treated with chemotherapy whereas in POLAR, 52% of patients received chemotherapy.
2. TAM is a cohort study whereas POLAR is a case-control design hence proportionally significantly more relapses are represented.
3. POLAR was initially designed to select cases and matched controls from a hospital cohort—matching on age at diagnosis, NPI category, type of endocrine therapy for 5 years, and whether chemotherapy was administered or not. Hence cases and controls (i.e., relapses and no relapses) have previously been matched.

There were 91 relapse events in POLAR and 84 controls (referred to as No Relapse). There were 43 early relapses (occurring up to five years after diagnosis) and 48 late relapses (occurring beyond 5 years). Two samples had insufficient material or failed at library construction. 175/194 samples passed sequence coverage thresholds. Baseline clinical and pathological characteristics of each study population are shown in Supplementary Table 2.

**Gene panel and capture probe design**. A meta-analysis of six published breast cancer sequencing studies[1,2,5,38–40] was performed and a multi-study mutation annotation format (MAF) file compiled that included all somatic exome mutations from the six studies (Supplementary Data 1). Kan et al.[39] targeted ~1500 coding genes whereas the other five used "whole-exome" capture approaches. Data were gathered from supplementary materials of each study or the TCGA data portal for that study[3]. Gene names were harmonized to official gene symbols using the NCBI Human Gene Info table (available at ftp://ftp.ncbi.nih.gov/gene/DATA/GEN-E_INFO/Mammalia/Homo_sapiens.gene_info.gz as of 4/3/15). The columns for protein change, mutation type (e.g., SNP, INS, DEL, etc), and mutation class (e.g., Missense, Silent, Frameshift, etc) were edited to use TCGA nomenclature. Finally, clinical subtype data was gathered and appended to the dataset where available. Intrinsic subtype was available for the Banerji et al.[1], Ellis et al.[5], and TCGA studies[3], while immunohistochemistry (IHC) results were available for the Kan et al.[39], Shah et al.[40], Stephens et al.[2], and TCGA publications. Genes were selected from this multi-study MAF file that were (1) recurrently mutated in 2% or more of cases or (2) recurrently mutated in 1% or more and determined to be druggable by their membership in greater than two druggable gene categories or having greater than two interactions with known anti-cancer drugs according to DGIdb[41]. Genes or their obvious family members were excluded if known to be problematic for variant calling (e.g., highly exonically variable). Unnamed genes (ENSG, KIAA, LOC) were also excluded. Genes from two additional large-scale sequencing studies were also considered. We included 15 of 16 genes reported as somatically altered by Chanock et al. from their own targeted sequencing of a 21 gene panel[42]. Three genes that were reported as deleted by the Curtis et al. study of copy number data in ~2000 patients were also included[15]. Further literature review was performed to include genes of known relevance to breast cancer (e.g., BRCA1, BRCA2, ERBB2, ESR1, and PRLR[43]), not meeting any of the criteria above. This resulted in a final list of 83 breast-cancer-related genes (Supplementary Table 1). Biotinylated 120 bp oligonucleotide probes were ordered from Integrated DNA Technologies (IDT, Coralville IA) and required to have no more than 50bp gaps between probes, and greater than 50% unique alignment to the reference genome and non-repetitive sequence content. These genes were targeted comprehensively with 337,144bp of unique coding exon sequences tiled by 3029 120 mer probes (a unique tiled space of 362,572 bp) (Supplementary Data 8).

**Library construction and sequencing**. TAM—Automated dual indexed libraries were constructed with 50–250 ng of genomic DNA utilizing the KAPA HTP Library Kit (KAPA Biosystems) on the SciClone NGS instrument (Perkin Elmer) targeting 250 bp inserts. Libraries with a starting input between 50–149 ng were enriched for 10 PCR cycles and libraries with a starting input between 150–250 ng were enriched for eight PCR cycles. A total of 92 libraries were pooled pre-capture

generating a 4.6 μg library pool. Library pools were hybridized with a custom set of probes (see above). The concentration of each captured library pool was determined through qPCR according to the manufacturer's protocol (KAPA Biosystems) to produce cluster counts appropriate for the Illumina HiSeq2000 platform. One lane of 2 × 100 sequence data generated an average of 375 Mb of data per sample. An average of 135.8× mean depth of coverage was achieved per sample.

MA12—Automated dual indexed libraries were constructed as above. A total of 90 libraries were pooled pre-capture generating a 5 μg library pool. Library pools were double hybridized with a custom set of probes (see above). The initial 4 h hybridization utilized the standard amount of probes followed by eight PCR cycles for enrichment. The second hybridization only required half of the standard probe concentration, followed by nine enrichment PCR cycles. The concentration of each captured library pool was determined as above to produce cluster counts appropriate for the Illumina HiSeq2500 platform. Two lanes of 2 × 125 sequence data generated an average of 668 Mb of data per sample. An average of 200× mean depth of coverage was achieved per sample.

POLAR—Automated dual indexed libraries were constructed with 25–250 ng of genomic DNA utilizing the KAPA HTP Library Kit (KAPA Biosystems) on the SciClone NGS instrument (Perkin Elmer) targeting 250 bp inserts. Libraries with a starting input between 25 and 149 ng were enriched for 11 PCR cycles and libraries with a starting input between 150 and 250 ng were enriched for eight PCR cycles. A total of 64 libraries were pooled pre-capture generating a 5 μg library pool. Library pools were hybridized with a custom set of probes (see above). The concentration of each captured library pool was determined as above to produce cluster counts appropriate for the Illumina HiSeq2500 platform. One lane of 2 × 125 sequence data generated an average of 750 Mb of data per sample. Approximately 75× mean depth of coverage was achieved.

**Primary variant calling and pre-filtering**. Variant calling was performed with the Genome Modeling System as previously described[44]. Specifically, sequence data were aligned to reference sequence build GRCh37-lite-build37 using bwa version 0.5.9[45] (params: -t 4 -q 5::), merged using picard version 1.46 [http://broadinstitute.github.io/picard], then deduplicated with picard version 1.46. SNVs were detected using the union of samtools version r963[46] and VarScan version 2.2.6[47] (params:–min-coverage 3–min-var-freq 0.08–p-value 0.10–strand-filter 1–map-quality 10) and processed through false-positive filter v1 (params:–bam-readcount-version 0.4–bam-readcount-min-base-quality 15–max-mm-qualsum-diff 100). Indels were detected using VarScan 2.2.6 (params: same as for SNVs) and filtered by false-indel version v1 (params:–bam-readcount-version 0.4–bam-read-count-min-base-quality 15–max-mm-qualsum-diff 100). Variants were annotated using Ensembl version 70. The initial variant set was expected to be dominated by germline variants and false positives. Therefore, a series of filters were applied. Variants were restricted to the regions of targeted genes. Variants were further excluded if found to have a global minor allele frequency in 1000 genomes (1092 individuals)[38], NHLBI exomes (SP6500SI-V2, 6503 individuals)[48], or ExAC[49] data sets greater than 0.1%. The 1000 genomes data (phase 1, release v3) were obtained directly from ftp://ftp.1000genomes.ebi.ac.uk and via dbSNP (v137). NHLBI Exomes were obtained from the Exome Variant Server (NHLBI GO Exome Sequencing Project, Seattle, WA, http://evs.gs.washington.edu/EVS/, downloaded 23 July 2012). Finally, a set of exome data for 151 unmatched normal blood samples[50] were processed through the identical variant calling pipeline as above. Any variant called in this study that was also observed in 10 or more of the unmatched normal samples was eliminated. This last filter was an attempt to eliminate common pipeline-related artifacts and common polymorphisms not present (or called) in 1000 genomes or NHLBI exomes. Variants from samples that failed coverage QC (Supplementary Data 2) were excluded.

**Variant post-filtering**. Inspection of the most highly recurrent variants identified several unlikely events in *FRG1B*, *MAP3K4*, and *MLL* that were mutated in more than 95% of samples. It was hypothesized that additional technical (alignment related) artifacts were causing these likely false positives. Rather than rely on variant calls in unrelated normals, a read counting approach was used to look for evidence of pipeline-related artifacts that might explain these variants. First, all variants were assessed for readcount support in a set of exome sequence data for 912 blood normal samples from the TCGA Breast Cancer project[3]. Variants were excluded from our list if at least 1% of TCGA samples had evidence of the same variant with at least three supporting reads, 1% VAF and 20× total coverage. Further inspection of recurrent variants in this newly filtered set identified two indels in the genes *RELN* and *FOXC1* that remained at suspiciously high quantities. Upon review of the TCGA normal exome panel it was discovered that the positions for these indels had very poor sequence coverage preventing identification of artifacts in these regions. Therefore, we further filtered against a set of 87 whole-genome sequenced blood normals also from the TCGA breast project. The same criteria were used as above except for a sample threshold of 2 out of 87. As a final step of quality control, each of these variants was manually reviewed. The entire pipeline was used on a set 16 tumors, sequenced with the same panel, for which matched normal variant calls (filtered and reviewed) were also available[51]. This identified a small number of remaining low-support false positives. Finally, a

Bayesian classifier was applied as previously described[52] to eliminate variants not predicted to be somatic with a binomial log likelihood ratio less than 10 for SNVs and less than six for indels.

**Detection of low frequency variants of known or potential clinical relevance**. To ensure that variants of known clinical relevance were not missed by automated variant calling approaches, a knowledge-based variant calling strategy was performed focused on the mutations in the Database of Curated Mutations[53]. This method scans directly for evidence of variants in raw alignment data using a database of mutations thought to be clinically or functionally relevant. At time of analysis the database included 413 total variant-disease entries for 35 genes from My Cancer Genome[54], Dienstmann et al.[55] and manual curation of the literature. These include well described variants of relevance in other cancers (e.g., *BRAF* V600E), as well as those specific to breast cancer such as *ESR1* ligand-binding domain[10,56] and *ERBB2* kinase domain mutations[57]. All samples were scanned for these known positions and potential variant calls made where at least five reads supported the variant with at least 1% VAF and 20× total coverage (Supplementary Data 3).

**Hotspot analysis**. The multi-study MAF file used for our original gene panel design (see above) was used as a reference to determine which variants in our study (if any) might represent novel sites of recurrent mutation, so called "hotspots". A recurrent mutation was considered novel if it was significantly enriched in our cohort by Fisher's exact test. Variants in both data sets were aggregated by amino acid position and counted. Each position was considered separately for the purposes of testing, and a minimum of three mutations in either cohort was required for inclusion.

**Univariate statistical analysis**. Patients were divided by mutation status or truncating mutation status for each gene. Fisher's exact and chi-squared tests were used for hotspot analysis, mutual exclusivity or co-occurrence, and other categorical clinical statistics (e.g., mutation status vs intrinsic subtype) as appropriate. Mutual exclusivity or co-occurrence tests were only performed for the top seven genes with the highest non-silent mutation rate. Paired *t*-tests or Mann–Whitney *U* tests (depending on normality of distributions tested via Kolmogorov–Smirnov test) were used for comparing group means for age, number of positive nodes, and tumor size. Kaplan–Meier survival analysis was performed for breast-cancer-specific survival (BCSS), relapse free survival (RFS; any regional or distant relapse), or overall survival (OS) with non-silent or truncating mutation status as a factor. Significant survival differences between the groups were determined by log-rank (Mantel–Cox) test. Survival tests were only performed if at least 15 patients had a non-silent mutation or at least eight patients had a truncating mutation. The Benjamini–Hochberg method was performed for multiple testing corrections to report the false discovery rate adjusted *p*-value (*q*-value).

**Multivariate statistical analysis**. The multivariate Cox proportional-hazard model was fitted to BCSS and RFS separately on the mutation status of a gene, besides classical clinical variables including node status (0 vs. 1~3 vs. 4+ positive), grade (1 or 2 vs. 3), and tumor size (1 vs. 2 vs. 3 vs. 4). To avoid the common issue of non-proportional-hazard and to clarify whether a gene's mutation shows early or late prognostic effect, the survival endpoints were each analyzed using the Cox model splitting at year 5. To adjust for multiplicity, the permutation procedure was performed. At each permutation, the survival and event outcome, together with clinic data, was permuted while mutation data of all genes were untouched but their original association with survival adjusted for clinical data was changed due to the permutation. The same Cox model was fitted on the permuted data to have its accompanied Wald test *p*-value derived. The permuted *p*-value associated with a gene's mutation was finally calculated as the proportion of permutations where at least one gene showed better *p*-value than the gene's original *p*-value for a survival endpoint. All statistical analyses were performed in the R statistical programming language with core, "survival" and "multtest" libraries. Genomic visualizations were created with ProteinPaint[58] and GenVisR[59].

**Data availability**. All mutation calls are made available as a MAF file with this publication. The raw sequence data from UBC-TAM patients are available in the database of Genotypes and Phenotypes (dbGaP) under accession number [dbGaP: phs001234]. Raw sequence data from MA12 and POLAR could not be deposited in public repository due to patient consent issues and complexities of institutional certification. However, these data are available from the authors (contact Obi Griffith and Matthew Ellis). Primary clinical outcome data for UBC-TAM and MA12 can be made available to qualified researchers through application to the Canadian Cancer Trials Group. Primary clinical outcome data for POLAR can be made available to qualified researchers through application to Mitch Dowsett at the Ralph Lauren Centre for Breast Cancer Research.

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

## Acknowledgements

Research reported in this publication was primarily supported by a Susan G. Komen Promise grant (PG12220321 to M.J.E.) and a Cancer Prevention and Research Institute of Texas (CPRIT) Recruitment of Established Investigators award (RR140033 to M.J.E.). M.J.E. is a McNair Medical Institute Investigator and a Susan G. Komen Scholar. The study was also supported by DOD BCRP award No. W81XWH-16-0538 to M.J.E. and E. C.C. S.M.K. was supported by a Komen CCR award (CCR16380599). The MA12 analysis was supported by research grants from Canadian Cancer Society Research Institute to the

NCIC Clinical Trials Group (021039 and 015469). O.L.G. was supported by the National Cancer Institute (NIH NCI K22CA188163 and NIH NCI U01CA209936).

## Author contributions

O.L.G., T.O.N., M.J.E., and E.R.M. designed the experiments; N.C.S., M.A., M.G., J. K., C. A.M., K.K., J.H., B.J.A., Z.L.S., K.C., R.K., C.F., L.C., J.E.S., S.D., V.M., D.E.L., R.S.F., S.L., and R.K.W. generated the sequencing data. T.O.N., B.Y., M.D., S.L., and D.V. orchestrated the sample pipeline, M.A., O.L.G., N.C.S., and J.K. prepared the figures and tables. O.L.G., N.C.S., M.A., J.L., J.K., and D.T. provided statistical analysis. S.M.K., R.B., and E. C.C. provided functional annotations. T.O.N. provided pathology analysis. M.J.E., N.C.S., M.A., and O.L.G. wrote the manuscript. E.R.M., T.O.N., and M.D., critically read and commented on the manuscript.

## Additional information

**Competing interests:** M.J.E., T.O.N., and E.R.M. report income on patents on the PAM50 intrinsic subtype algorithm. M.J.E. reports ownership in Bioclassifier LLC that licenses PAM50 patents to Nanostring for the Prosigna breast cancer prognostic test. Commercial platforms and algorithms were not used in the analyses reported in this paper. The remaining authors declare no competing interests.

