## [Peer Review File · Nature Communications]

Reviewers' comments:

Reviewer #1 (Remarks to the Author):

Authors perform a NGS panel of 83 genes on a large combined dataset (n=1128) of primary breast cancers (involving 2 clinical trial datasets) with patients spanning a large range of ages, stages, length of follow-up and treatment.

Data presented is largely descriptive but adds to current data and publications re Metabric and TCGA given that these breast cancer samples were highly selected, heterogenous treatments depth of sequencing was not high and follow up for TCGA is not mature. Prognostic data presented on NF1, PIK3R1 (DDR1) mutations unfortunately involve small numbers of patients but would be useful for the field for future validation for these low frequency mutations. Co-existence of PIK3CA and MAP3K1 and good outcome seem to be only in the post-menopausal population. Interesting data in Fig 8 as well as on CFBF. Unfortunately no conclusions about treatment prediction can be taken.

For some of the lower frequency significantly prognostic mutations, I wonder if orthogonal validation may be useful, as these have not been observed before.

Were the HER2-enriched actually HER2 positive? Were these included in the somatic mutation and prognostic analyses? (Figure 2b)

Were the TAM and MA12 gene alteration validations in the Metabric performed in post-menopausal TAM treated only and pre-menopausal patients respectively? Given the little overlap between the TAM and MA12 datasets do the authors speculate that the genes relate to the patient populations (menopausal status?) was there other differences in these datasets by grade, tumor size etc.

Copy number alterations are key drivers of proliferation and prognosis in HR-positive breast cancer and have not been described here. This is worthy of note in the discussion – in addition to consideration that mutations may have varied prognostic effects depending on their context (both intrinsic subtype and genetic context). This could be done presumably using existing data. I would consider this to be a major weakness of the paper.

Other minor

- I like to see numbers of patients/events for Figures 3,4,5, 6 forest plots
- Was the definition of ER positivity different by dataset (ie 1% vs 10%)
- Was the PAM50 analyses in MA12 done only in ER+ (IHC) patients?
- Discussion should acknowledge the heterogeneity in datasets and treatments
- Comment should be made re rate of germline potential genes BRCA etc in the MA12 population-
Line 215 – regarding FS/NS and non-silent mutations in ATM in younger women in MA12 – are these considered definitely somatic, or are any of these potentially germline mutations? (could be explored by looking at allele frequencies and Clinvar annotations) line 306 as well refers to this
- Line 213 – regarding FS/NS mutations in GATA3 – it is noted that they have a strong association between luminal B status but I can only see results for the MA12 study . Was this consistent in the UBC-TAM study also?
- Line 252 – Regarding the survival endpoint for the METABRIC study – “disease-specific outcome” – could the authors clarify what this is? How does this compare with BCSS and OS, the endpoints from UBC-TAM and MA12?
- Line 300 – the term “mutation rate”– suggest changing to mutational frequency.
- Line 300 - The authors note different mutational frequencies compared with TCGA – do the authors feel that patient selection might also be a possible reason rather than sequencing

techniques alone?

- Figure 4b needs endpoint ?DFS etc like Figure 3b

Presumably this data will be publicly available post publication.

Reviewer #2 (Remarks to the Author):

The manuscript by Griffith et al. examines the mutational profile in the archival samples from three clinical trials of ER+ breast cancer patients undergoing adjuvant treatment. The significant correlations were then validated against the METABRIC study. The results show that NF1, PIK3R1, MAP3K1, PIK3CA and TP53 are likely prognostic drivers with DDR1, PRKDC and XBP1 showing promise. Though the manuscript did not present new biomarkers, it represented important work for the clinical annotation of somatic mutations. The two stage design with validation in the METABRIC cohort provides reassurance of relative importance. The results will be used by many in the clinical interpretation of such somatic mutations.

The manuscript however, could be written in a more clear manner. For example, the analysis primarily was for the TAM and the MA12 cohorts but it was difficult to see what happened to the data from the POLARIS cohort. The most important omission, however, is a much better description of the functional coverage of their capture array. As with any targeted genomic sequencing, the exons covered for how many genes (and which they are) is important for the users of this data. This information is tucked away in supplemental table 1 left to the cognoscenti. But the less technical reader will be left with the message that these were the only important mutation in a comprehensive genomic scan (which is not the case). It will be important to describe in the text how many genes were studied and what depth of coverage was performed. This also could be brought into relief with results of other such studies with different limited sequencing panels in the clinical domain.

Lastly, it would be helpful to the clinical reader if the authors would talk about the actionability of these significant somatic mutations.

We would like to thank the editors for inviting us to resubmit our revised manuscript “The prognostic effects of somatic mutations in ER-positive breast cancer” after addressing the reviewers’ comments. We greatly appreciate the feedback from the reviewers, which we believe has substantially strengthened the impact of our analysis and results. Below, *italicized text* includes comments directly from reviewers’ feedback followed by **plain blue text** describing how we addressed these specific comments in our revisions. Added or modified portions of the manuscript itself are **highlighted in red**. Figures within this response are referred to as “Figure RX” to prevent confusion with figures in the manuscript.

Reviewers' comments:

Reviewer #1 (Remarks to the Author):

Authors perform a NGS panel of 83 genes on a large combined dataset (n=1128) of primary breast cancers (involving 2 clinical trial datasets) with patients spanning a large range of ages, stages, length of follow-up and treatment.

Data presented is largely descriptive but adds to current data and publications re Metabric and TCGA given that these breast cancer samples were highly selected, heterogenous treatments depth of sequencing was not high and follow up for TCGA is not mature. Prognostic data presented on NF1, PIK3R1 (DDR1) mutations unfortunately involve small numbers of patients but would be useful for the field for future validation for these low frequency mutations. Co-existence of PIK3CA and MAP3K1 and good outcome seem to be only in the post-menopausal population. Interesting data in Fig 8 as well as on CFBF. Unfortunately no conclusions about treatment prediction can be taken.

By “treatment prediction”, we assume that the reviewer meant endocrine treatment prediction. Endocrine treatment in UBC Tam was uniform but not randomized. In MA12 the use of tamoxifen was randomized, but the numbers were too small to examine treatment interactions. A comment regarding this weakness has been added to the discussion.

Addition to the beginning of the discussion

The strength of this investigation includes the prolonged follow up, controlled adjuvant treatment and the relatively large number of genes and patients studied. Weaknesses include the lack of treatment prediction because endocrine treatment in UBC Tam was uniform but not randomized. In MA12 the use of tamoxifen was randomized, but the numbers were too small to examine treatment interactions.

For some of the lower frequency significantly prognostic mutations, I wonder if orthogonal validation may be useful, as these have not been observed before.

We were uncertain whether the reviewer was referring to orthogonal validation of the prognostic associations, functional validation of the putatively prognostic mutations, or sequence validation

of the mutations themselves. For the mutations themselves, we argue that orthogonal sequence validation is not necessary. Given the high coverage, extensive filtering, and manual review of reported variants we can expect that the validation rate of simple SNVs (e.g., by Sanger sequencing) will be very high and therefore unnecessarily redundant based on past reports by our group and others (PMID:25960255). For validation of prognostic associations we did report orthogonal validation using the METABRIC dataset, the largest (and most appropriate) validation existing dataset for this purpose. And, indeed for some of the key low frequency significantly prognostic mutations (e.g., NF1 in UBC-TAM, PIK3R1 in MA12) we were able to validate these prognostic associations. Finally, functional validation of some of the key low-frequency findings (e.g., NF1 and DDR1) are also underway but are beyond the scope of this paper. We have modified the end of the discussion to clarify that a more complete understanding of the significance of low frequency prognostic mutations awaits additional future validation and functional studies.

Addition to the manuscript

In the meantime, functional studies should be pursued to understand the biological effects of low frequency somatic mutations, prioritizing these studies according to whether the mutations are driving an adverse prognostic effect and whether their disruption creates a therapeutic vulnerability.

Were the HER2-enriched actually HER2 positive? Were these included in the somatic mutation and prognostic analyses? (Figure 2b)

To address this question we show distribution of Her2 IHC/FISH (+ve/-ve) status and mutations in HER2 across UBC-TAM+MA12 cohort categorized by intrinsic subtypes (Figure R1) and have added relevant text to the results section. It is worth reporting that, as expected, we see an enrichment of alterations which can lead to HER2-overexpression (HER2 +ve by FISH/IHC AND/OR HER2 Mutations) in cases categorized as HER2-E subtype (Figure R2, Fisher's exact test $p < 0.0001$, shown below).

Figure R1. Her2 IHC/FISH (+ve/-ve) status and mutations in HER2 across UBC-TAM+MA12 cohort categorized by intrinsic subtypes

PAM50	HER2 -ve		HER2 +ve		HER2 Unknown	
	HER2 WT	HER2 Mut	HER2 WT	HER2 Mut	HER2 WT	HER2 Mut
LuminalA	324	11	18	1	6	1
LuminalB	230	8	28	0	4	0
Basal	14	1	1	1	0	0
HER2-E	47	4	38	0	0	0

Figure R2. Enrichment of HER2 alterations (mutation, +ve by FISH/IHC) in HER2e subtype versus other subtypes.

Addition to manuscript:

As expected, patients with the HER2-E intrinsic subtype were enriched for HER2+ve status compared to other subtypes (Fisher's exact test $p < 0.0001$). Of interest, in the HER2-enriched group there were 51 tumors that were not HER2 amplified and of these 4 were HER2 mutant (~8%), indicating that HER2 mutation could be an occasional explanation for a HER2-E subtype assignment in the absence of HER2 amplification.

This question actually led to another finding that we had not previously considered, namely the following issue with NF1 which we decided to include since the HER2-E HER2 negative group are somewhat enigmatic.

Addition to manuscript:

For NF1 FS/NS mutations, there was also a statistically significant association with the HER2-E subtype ($P = 0.002$) (supplementary Figure 7B, also supplementary data 5). Notably NF1 non-silent mutations were enriched in the HER2-E non-HER2 amplified subgroup, where they were present in 8/51 cases (16%). Compared to the frequency in all other subtypes 12/582 (2%), this enrichment was significant (Fishers exact test $p < 0.0001$) (Supplementary 7A right panel). This association could be reproduced in the METABRIC data with an NF1 non-silent mutation incidence in the HER2-E non HER2 amplified group of 8/80 (10%) versus 35/1283 (3%) in the rest of the subtypes ($p = 0.003$) (Supplementary 7A right panel).

We also added the following comment in the discussion.

The association with the HER2-E, non-HER2 amplified subset with non-synonymous NF1 mutations was observed in both the discovery and validation (METABRIC) data sets. It is a logical proposition that mutations that activate RAS, like NF1 mutation, could create a tumor with a similar transcriptional phenotype as some HER2 amplified breast cancers.

Were the TAM and MA12 gene alteration validations in the Metabric performed in post-menopausal TAM treated only and pre-menopausal patients respectively? Given the little overlap between the TAM and MA12 datasets do the authors speculate that the genes relate to the patient populations (menopausal status?) was there other differences in these datasets by grade, tumor size etc.

As a validation for the TAM candidates, we added a new supplementary table, and related text, reporting univariate and multivariate analysis on candidate genes restricting METABRIC patient samples to those who received Hormone therapy and were postmenopausal (Supplementary Table 4). We see hazard ratios very similar to those reported in Figure 5B and hence, the conclusions of the analysis remain intact. A similar analysis could not be performed for premenopausal hormone-therapy treated women for validation of MA12 candidates because of lack of samples qualifying that criteria in METABRIC. This further illustrates the uniqueness of MA12 patient cohort.

Supplementary Table 4. Univariate and multivariate analysis of TAM candidates in hormone-therapy-treated post-menopausal METABRIC subjects.

Anaysis	Gene	VariationT	HR	P	lower	upper
1UVA	ARID1B	non-silent	0.823	0.68	0.3323	2.041
MVA	ARID1B	non-silent	0.729	0.49	0.294	1.807
1UVA	ERBB3	non-silent	0.8846	0.787	0.36	2.156
MVA	ERBB3	non-silent	0.8397	0.701	0.3435	2.053
1UVA	MAP3K1	non-silent	0.751	0.239	0.466	1.211
MVA	MAP3K1	non-silent	0.771	0.294	0.4751	1.253
1UVA	NF1	FS/NS	2.44	0.049	1	5.94
MVA	NF1	FS/NS	3.38	0.008	1.38	8.317
1UVA	PIK3CA	non-silent	1.03968	0.7954	0.7747	1.395
MVA	PIK3CA	non-silent	1.02699	0.86327	0.7584	1.391
1UVA	TP53	non-silent	2.1968	1.62E-07	1.624	2.972
MVA	TP53	non-silent	2.0829	6.63E-06	1.5137	2.866
clinical	Tumor Gra	Clinical	1.5277	0.005697	1.131	2.063
clinical	Node Posi	Clinical	3.032	9.14E-09	2.077	4.426
clinical	Tumor Size	Clinical	4.076	1.20E-07	2.423	6.858

Addition to manuscript:

In order to maintain coherence in discovery and validation patient cohorts, a similar analysis was carried out restricting the patient pool to postmenopausal patients only. No significant variation in hazard ratio for candidate genes where observed (Supplementary Table 4).

Copy number alterations are key drivers of proliferation and prognosis in HR-positive breast cancer and have not been described here. This is worthy of note in the discussion – in addition to consideration that mutations may have varied prognostic effects depending on their context (both intrinsic subtype and genetic context). This could be done presumably using existing data. I would consider this to be a major weakness of the paper.

To take into account the role of CNV and “genetic context” in our analysis of mutations associated with prognosis, we have added 2 dimensions to our analysis.

- 1) We now add information on copy number variation proportions to figures 5 and 6 for the candidate gene mutated cases. Here, we use 4 broad classifications of CNVs (standard for CBioPortal): Deep Deletion, Shallow Deletion, Amplification, and Gain (low level gains) for both the validation analyses (See modified Figures 5 and 6).
- 2) We gauge the effect of occurrence of MYC, FGFR1, CCND1 and ERBB2 amplifications, which have been associated with ER-positive breast cancer in earlier reports {ref PMID10706127, PMC3092228, PMID15574759 }, on the prognostic association of the candidate genes for both UBC-TAM and MA12 candidate genes when tested in the METABRIC cohort (for which CNV data are available). We see hazard ratios very similar to the ones reported in Figure 5B and 6B and hence, the conclusions of the analysis remain intact. We report this multivariate analysis in Supplementary Table 5.

Supplementary Table 5. Analysis of effect of considering CNV status of selected amplification status on mutation-prognosis associations

Dataset	Analysis	Analysis_type	Gene	VariationType	HR	P	lower	upper
METABRIC_forTAM	MVA:AMP-ERBB2,FGFR1,CCND1,MYC	BCSSS	ARID1B	non-silent	0.727	0.485087	0.297	1.779
METABRIC_forTAM	MVA:AMP-ERBB2,FGFR1,CCND1,MYC	BCSSS	ERBB3	non-silent	0.839	0.649144	0.3938	1.787
METABRIC_forTAM	MVA:AMP-ERBB2,FGFR1,CCND1,MYC	BCSSS	MAP3K1	non-silent	0.6124	0.04324	0.3807	0.9852
METABRIC_forTAM	MVA:AMP-ERBB2,FGFR1,CCND1,MYC	BCSSS	NF1	Truncating	2.1325	0.069195	0.9422	4.826
METABRIC_forTAM	MVA:AMP-ERBB2,FGFR1,CCND1,MYC	BCSSS	PIK3CA	non-silent	1.07086	0.609028	0.8237	1.392
METABRIC_forTAM	MVA:AMP-ERBB2,FGFR1,CCND1,MYC	BCSSS	TP53	non-silent	2.0428	6.71E-07	1.5413	2.708
METABRIC_forMA12	MVA:AMP-ERBB2,FGFR1,CCND1,MYC	OS	ERBB2	nonsilent	1.38267	0.15531	0.8844	2.162
METABRIC_forMA12	MVA:AMP-ERBB2,FGFR1,CCND1,MYC	OS	ERBB4	nonsilent	0.74375	0.4392	0.3513	1.575
METABRIC_forMA12	MVA:AMP-ERBB2,FGFR1,CCND1,MYC	OS	JAK1	MS	1.418429	0.19711	0.8339	2.413
METABRIC_forMA12	MVA:AMP-ERBB2,FGFR1,CCND1,MYC	OS	PIK3R1	nonsilent	1.83177	0.00955	1.159	2.895
METABRIC_forMA12	MVA:AMP-ERBB2,FGFR1,CCND1,MYC	OS	PIK3R1	truncating	2.24241	0.01161	1.1977	4.198
METABRIC_forMA12	MVA:AMP-ERBB2,FGFR1,CCND1,MYC	OS	RB1	nonsilent	1.19135	0.4926	0.7225	1.964

Additions to manuscript:

Copy number aberrations and chromosomal instability have been associated with prognosis across multiple cancer types, including ER-positive (ER+) breast cancer^{16, 18, 19}. To gauge the confounding nature of commonly amplified genes in breast cancer, we further performed multivariate analysis on the candidate genes with cases of amplification of MYC, FGFR1, CCND1 and ERBB2 (Supplementary Table 5). We did not observe a significant change in the hazard ratio reported in Figure 5B and 6B).

A limitation of this study is that the mutation datasets we generated for UBC-TAM and MA12 cohorts lack comprehensive assessment of copy number signatures that have been associated with prognosis in ER+ breast cancer^{16, 18, 19}. While multivariate analysis considering key CNVs did not appear to affect our prognostic associations, future studies may be needed to completely understand the interplay between simple and large-scale variation for prognostic prediction.

Other minor

- I like to see numbers of patients/events for Figures 3,4,5,6 forest plots

The figures have been updated with number of cases/patients.

- *Was the definition of ER positivity different by dataset (ie 1% vs 10%)*

We have clarified in the study cohort descriptions (See Results) that in all cases 1% of cells was considered positive for ER/HR/PR status.

- *Was the PAM50 analyses in MA12 done only in ER+ (IHC) patients?*

Yes. The PAM50 analysis was limited to the HR+ subset of patients as this was relevant question in this trial. This has been clarified in the text.

- *Discussion should acknowledge the heterogeneity in datasets and treatments*

We acknowledge the heterogeneity in the introduction when we introduce the datasets but we will have now revisited this important caveat in the discussion.

Addition to manuscript:

Another limitation to this study was the heterogeneity in the datasets in terms of age, treatment, and other factors that limited direct comparison and made validation with METABRIC somewhat challenging. The collection of sufficiently large, uniformly treated populations with long-term follow-up for discovery and validation remains a challenge that must be addressed to fully characterize the prognostic significance of somatic mutations, especially low-frequency mutations.

- *Comment should be made re rate of germline potential genes BRCA etc in the MA12 population- Line 215 – regarding FS/NS and non-silent mutations in ATM in younger women in MA12 – are these considered definitely somatic, or are any of these potentially germline mutations? (could be explored by looking at allele frequencies and Clinvar annotations) line 306 as well refers to this*

This is an excellent suggestion. We have extended the analysis of BRCA1/2 potentially germline/pathogenic variants to MA12. We have also extended germline analysis to ATM for both UBC-TAM and MA12 (Supplementary Data 9). We do show that both together (and individually) these genes have significantly higher VAFs suggesting increased numbers of de novo germline mutations in these genes compared to other genes (that are more consistently driven by only somatic mutations). The same analysis procedure used for BRCA1/2 was used for ATM. Pathogenicity was determined by querying each variant in ClinVar and LOVD. We also tested for BRCA1 VAF differences in the MA12 HR- cohort compared to the VAF of all other genes (including BRCA2 and ATM) and saw BRCA1 VAF as significantly higher (p-value=0.00024). These results were not included in the manuscript as the focus of this paper is on HR+ patients.

Addition to manuscript:

Of the 117 non-silent *BRCA1/2* mutations observed (from 110/1128 patients across all 3 cohorts; 7 patients had two hits) 74 were observed at a VAF greater than 40% and 31 were greater 60%. Additionally, of the 61 non-silent *ATM* mutations (from 58/1128 samples; 3 samples had 2 hits) 39 had VAF greater than 40% and 18 had VAF greater than 60 (Supplementary Data 9). Variants with VAFs this high are less likely to be somatic given the general expectation of impure tumor samples and heterozygous mutations. Indeed, the VAFs for *BRCA1/2* and *ATM* non-silent mutations (mean=46.0%) were significantly higher than for other genes (mean=36.7%, $p=5.92e-09$). Even when considered separately, the VAFs for *BRCA1* (mean=46.6%), *BRCA2* (mean=43.8%) and *ATM* (mean=48.2%) were significantly higher than the other genes ($p=0.002$, $p=0.0015$, and $5.27e-5$ respectively). Among the *BRCA1/2* variants, there were 8 known pathogenic (ENIGMA expert reviewed) mutations according to a search of the BRCA Exchange database (<http://brcaexchange.org>, Nov 12, 2017) and another 37 assumed pathogenic (FS/NS) mutations. Of the remaining, 4 were benign according to expert review (ENIGMA), and 8 benign, 15 likely benign and 45 variants of unknown significance according to all public sources. Out of the 61 *ATM* variants queried in ClinVar, 4 were designated as pathogenic, 3 were pathogenic/likely pathogenic, and 2 were likely pathogenic. Another 7 were frameshift mutations and assumed pathogenic. Additionally, 23 variants had uncertain significance, 8 variants had conflicting interpretations of pathogenicity (any combination of benign, likely benign, or uncertain significance), and the remaining 14 variants had no data. *ATM* variants were also queried in the Leiden Open Variation Database (LOVD)²⁴, which identified 1 variant that affects function (designated as likely pathogenic by ClinVar), 10 variants with unknown effect, and 1 variant that probably does not affect function (uncertain significance in ClinVar). The remaining variants had no data in LOVD. Given these complexities the prognostic effects of *BRCA1/2* and *ATM* remain unresolved, however attention should clearly be paid to therapeutic strategies for these patients. The *ATM* findings deserve a particular highlight because of the younger age/luminal B association and the current lack of studies devoted to this population.

• *Line 213 – regarding FS/NS mutations in GATA3 – it is noted that they have a strong association between luminal B status but I can only see results for the MA12 study. Was this consistent in the UBC-TAM study also?*

No, this was non-significant in the TAM data. However, the number of luminalB patients is much lower in TAM, possibly affecting our power to detect such an association. The subtype-mutation associations are generally consistent between TAM and MA12 (see TP53 and PIK3CA) but for lower frequency mutations it is harder to assess. We have clarified this in the results.

• *Line 252 – Regarding the survival endpoint for the METABRIC study – “disease-specific outcome” – could the authors clarify what this is? How does this compare with BCSS and OS, the endpoints from UBC-TAM and MA12?*

We used BCSS for TAM which is equivalent to DSS used in METABRIC validation of TAM findings. We used OS for MA12 and OS for METABRIC validation of MA12 findings. This has been clarified in the text of the manuscript and a consistent term BCSS was used instead of DSS throughout.

- *Line 300 – the term “mutation rate”– suggest changing to mutational frequency.*

This has been changed throughout the manuscript as suggested.

- *Line 300 - The authors note different mutational frequencies compared with TCGA – do the authors feel that patient selection might also be a possible reason rather than sequencing techniques alone?*

This is quite possible. We compared our overall mutation frequencies (UBC-TAM, MA12, and POLAR) to the TCGA ER+ subset only. However, it is true that our patients are more clinically homogenous than TCGA, and this is a plausible explanation for differences in frequencies, in addition to technology differences. We also note that our frequencies are much closer to those reported in the METABRIC study. The discussion/results have been modified to reflect these points.

Addition to manuscript:

Results: Considering METABRIC ER+ patients, the most recurrently mutated genes were PIK3CA (~46%), TP53 (~21%), GATA3, MLL3, CDH1, and MAP3K1 (all ~12-14%) demonstrating slightly higher but very similar frequencies.

Discussion: Differences in patient populations may also be a factor. Frequencies were much closer to reported values for METABRIC.

- *Figure 4b needs endpoint ?DFS etc like Figure 3b*

We have added “OS” to Fig 4B as suggested.

Presumably this data will be publicly available post publication.

Yes. To the extent that this is possible we are very eager to share the data. A ‘Data Availability’ section has been added to the manuscript explaining how to obtain the various datasets.

Addition to manuscript:

All mutation calls are made available as a MAF file with this publication. The raw sequence data from UBC-TAM patients are available in the database of Genotypes and Phenotypes (dbGaP)

under accession number [dbGAP:phsxxxx.x]. Raw sequence data from MA12 and POLAR could not be deposited in public repository due to patient consent issues and complexities of institutional certification. However, these data are available from the authors (contact Obi Griffith and Matthew Ellis). Primary clinical outcome data for UBC-TAM and MA12 can be made available to qualified researchers through application to the Canadian Cancer Trials Group. Primary clinical outcome data for POLAR can be made available to qualified researchers through application to Mitch Dowsett at the Ralph Lauren Centre for Breast Cancer Research.

Reviewer #2 (Remarks to the Author):

The manuscript by Griffith et al. examines the mutational profile in the archival samples from three clinical trials of ER+ breast cancer patients undergoing adjuvant treatment. The significant correlations were then validated against the METABRIC study. The results show that NF1, PIK3R1, MAP3K1, PIK3CA and TP53 are likely prognostic drivers with DDR1, PRKDC and XBP1 showing promise. Though the manuscript did not present new biomarkers, it represented important work for the clinical annotation of somatic mutations. The two stage design with validation in the METABRIC cohort provides reassurance of relative importance. The results will be used by many in the clinical interpretation of such somatic mutations.

We thank the reviewer for their words of encouragement.

The manuscript however, could be written in a more clear manner. For example, the analysis primarily was for the TAM and the MA12 cohorts but it was difficult to see what happened to the data from the POLARIS cohort.

The POLAR cohort was a case-control design that did not allow for prognostic (survival analysis) in the same way that MA12 and UBC-TAM data did. For that reason we focused primarily on the latter two cohorts. However, we did include POLAR for non-survival-based analyses such as hotspot and mutation landscape analysis. The abstract and discussion therefore are focused primarily on UBC-TAM and MA12. This has been clarified/emphasized in the discussion.

The most important omission, however, is a much better description of the functional coverage of their capture array. As with any targeted genomic sequencing, the exons covered for how many genes (and which they are) is important for the users of this data. This information is tucked away in supplemental table 1 left to the cognoscenti. But the less technical reader will be left with the message that these were the only important mutation in a comprehensive genomic scan (which is not the case). It will be important to describe in the text how many genes were studied and what depth of coverage was performed. This also could be brought into relief with results of other such studies with different limited sequencing panels in the clinical domain.

For space reasons we were forced to move much of the details of the gene panel sequencing to the supplementary methods, tables and figures. Indeed, the supplementary methods includes a large section on ‘Gene panel and capture probe design’. Supplementary figures 1-5 are devoted exclusively to describing the functional coverage of the capture array, including percent of exon bases covered at different minimum coverage cutoffs, percent alignment rates, mean depth and percent duplicates and their relation to library input and sample age, overall distributions of mean depth and total bases sequenced, and finally gene level coverage. We also touch on this issue in Supplementary Figures 9-10 where we show how TCGA data had poor coverage for CFB splice sites whereas our panel had good coverage allowing us to identify a novel splice site hotspot mutation. We do try to make it clear, in the abstract, that we subjected samples to “targeted” sequencing and in the introduction that “83 genes were chosen for analysis” and “83 genes [were] investigated in the study described here”. The fact that this study describes an 83 gene panel (and not a comprehensive genomic scan) are mentioned an additional 5 times in the manuscript and also in main Figures 3 and 4. We also describe in the results the coverage criteria for passing samples and give the overall mean coverage, referring the reader to the supplement for more details.

To further clarify these issues we have added explicit mention of the 83-gene panel size to the abstract of the paper. Additional sequence coverage details have been added to the results. And, we also now summarize coverage data in the main Figure 1 (top side bar) to show our very comprehensive coverage of the targeted exon space for all samples passing quality criteria and included in the study.

Lastly, it would be helpful to the clinical reader if the authors would talk about the actionability of these significant somatic mutations.

We have added a caveat to the paper about the difficulty of using this mutation data to directly predict response tamoxifen (see response to reviewer #1). We previously included a section analyzing in detail the potential significance of mutations in uncommon targetable kinases. The section on potential clinical significance of germline cancer predisposition mutations has now been significantly expanded (see response to reviewer #1). We now talk about the potential clinical relevance of ERBB2 kinase domain mutations with regards to directed therapy such as neratinib. Finally, in the discussion we do talk about the potential future clinical significance of NF1, DDR1, and other novel prognostic association mutations identified here. However, future functional studies and drug development are needed before these mutations are actionable.

REVIEWERS' COMMENTS:

Reviewer #1 (Remarks to the Author):

comments are satisfactorily addressed.
will be an important paper for the field.

Reviewer #2 (Remarks to the Author):

The authors answered my concerns. One suggestion for a better final manuscript would be to add a sentence or two as to why this work adds to the knowledge base: e.g. long follow up; a comprehensive overview of published datasets that examines the highest number of patients; etc.